# Neural Population Geometry Reveals the Role of Stochasticity in Robust Perception

**Joel Dapello**[*,1,2,3], **Jenelle Feather**[*,1,2,4], **Hang Le**[*,1], **Tiago Marques**[1,2,4]
**David D. Cox**[5], **Josh H. McDermott**[1,2,4,6], **James J. DiCarlo**[1,2,4], **SueYeon Chung**[1,7,8]

[*]Equal contribution, ordered alphabetically
[1]Department of Brain and Cognitive Sciences, Massachusetts Institute of Technology
[2]McGovern Institute for Brain Research, Massachusetts Institute of Technology
[3]School of Engineering and Applied Sciences, Harvard University
[4]Center for Brains, Minds and Machines, Massachusetts Institute of Technology
[5]MIT-IBM Watson AI Lab
[6]Speech and Hearing Bioscience and Technology, Harvard University
[7]Center for Theoretical Neuroscience, Columbia University
[8]Zuckerman Institute, Columbia Univeristy

## Abstract

Adversarial examples are often cited by neuroscientists and machine learning researchers as an example of how computational models diverge from biological sensory systems. Recent work has proposed adding biologically-inspired components to visual neural networks as a way to improve their adversarial robustness. One surprisingly effective component for reducing adversarial vulnerability is response stochasticity, like that exhibited by biological neurons. Here, using recently developed geometrical techniques from computational neuroscience, we investigate how adversarial perturbations influence the internal representations of standard, adversarially trained, and biologically-inspired stochastic networks. We find distinct geometric signatures for each type of network, revealing different mechanisms for achieving robust representations. Next, we generalize these results to the auditory domain, showing that neural stochasticity also makes auditory models more robust to adversarial perturbations. Geometric analysis of the stochastic networks reveals overlap between representations of clean and adversarially perturbed stimuli, and quantitatively demonstrates that competing geometric effects of stochasticity mediate a tradeoff between adversarial and clean performance. Our results shed light on the strategies of robust perception utilized by adversarially trained and stochastic networks, and help explain how stochasticity may be beneficial to machine and biological computation.[1]

## 1 Introduction

In recent years, artificial neural networks (ANNs) have come to dominate both visual object recognition and auditory recognition tasks [1, 2, 3], establishing them as leading candidate models for several domains of human perception [4, 5, 6, 7]. However, they still exhibit many non-human-like traits [8, 9]. One such failure is in the existence of adversarial perturbations – small changes to stimuli explicitly crafted to fool a model that remain imperceptible to humans [10, 11, 12] – which demonstrate the fragility of some ANNs as models of biological perception.

---

[1]See https://github.com/chung-neuroai-lab/adversarial-manifolds for accompanying code.

35th Conference on Neural Information Processing Systems (NeurIPS 2021).

Recently, Dapello, Marques et al. discovered that one such method, known as adversarial training [13], not only reduces the network's adversarial vulnerability but also yields network representations that are more similar to those in the primate primary visual cortex [14]. Motivated by this result, the authors developed VOneNets, a class of networks that simulate the primate primary visual cortex at the front of a convolutional neural network, and show improved robustness to adversarial attacks with no adversarial training. However, a number of questions remain unanswered – in particular, while both adversarially trained networks and VOneNets have improved adversarial robustness and also greater similarity to the primate primary visual cortex, it is unclear how VOneNets achieve robustness, and in particular if the mechanism of robustness is similar to that induced by adversarial training.

A key component of robustness in VOneNets is the inclusion of stochastic representations during both training and inference, a feature inspired by biological sensory neurons which exhibit trial-to-trial variability across presentations of the same stimulus [15]. The implications of this stochasticity for information processing are open questions in neuroscience [16, 17, 18, 19]. Pinpointing how this representational stochasticity contributes to robustness in VOneNets could drive further developments in the mechanisms of robust perception.

Here we use recently developed manifold analysis techniques from computational neuroscience [20] to look beyond accuracy and investigate the internal neural population geometry [21] of standard, adversarially trained, and biologically-inspired stochastic networks in response to clean and adversarially perturbed examples in both visual and auditory domains. We present several key findings:

- Using manifold analysis, we demonstrate that standard, adversarially trained, and stochastic networks each have distinct geometric signatures in response to clean and adversarially perturbed stimuli, shedding light on varied robustness mechanisms.

- We demonstrate the generality of our findings by translating the results to a novel biologically-inspired auditory ANN, StochCochResNet50, that includes stochastic responses. Stochasticity makes auditory networks more robust to adversarial perturbations, and the underlying neural population geometry is largely consistent with that in vision networks.

- Analysis of stochastic networks reveals a protective overlap between the representations of adversarial examples and clean stimuli. We quantitatively survey the stochasticity conditions leading to the overlap, and map a competing geometric effect mediating a trade-off between clean and adversarial performance.

## 2 Related work

Previous work in machine learning has reported that additive noise can improve the adversarial robustness of a model: Liu et al. [22] used random noise applied to pixels and intermediate layers to improve adversarial robustness, and Cohen et al. [23] demonstrated a method to transform a model with Gaussian noise in the pixel space to one with certified robustness to attacks of a given strength. Unlike VOneNets, both [22] and [23] rely on using multiple inference passes with different noise samples to improve robustness. Further, while both of these works suggest new defenses, none analyze the properties of stochastic representations that lead to robustness.

The need to understand internal mechanisms of biological and artificial neural networks gave rise to the field of neural population geometry [24, 21], an line of work exploring geometric properties of high-dimensional neural representations. To capture the complexity of neural representations in ANNs, various geometric analyses have been proposed, including representation similarity analysis [25], geodesics [26], curvature [27], and intrinsic dimensionality [28]. Another popular approach is through supervised linear probes [29], i.e., linear classifiers trained on top of these representations. However, recent work has discussed how such analyses with linear classifiers are limited [30], and that more structural studies are needed for investigating the internal layers of networks [31]. A recent theoretical development based on replica theory in statistical physics [32, 20, 33] provides a solution to this, by formally connecting the geometry and linearly decodable information embedded in neural populations. By using this approach, our study situates the analysis of robustness mechanisms into the developing field of neural population geometry [24, 21].

# 3 Methods and experimental setup

## 3.1 Replica-based manifold analysis

In this paper, we use replica mean-field theoretic manifold analysis (MFTMA, hereafter) [20, 34, 33], which formally connects the linear decodability of object manifolds, defined as a set of stimulus-evoked representations in neural state space [35], to their geometrical properties. Here we provide a brief description of the key quantities (See SM 1 and [20] for a complete treatment of MFTMA). Using this framework, we analyze the size, shape, and distribution of object manifolds as they are transformed throughout the network, gaining insight into the neural population geometry underlying performance of the classification task. Specifically, given $P$ object manifolds (measured as feature point clouds) in $N$ dimensions, MFTMA returns capacity (a measure of linear separability) and the manifold dimensions, radii, and center correlations associated with capacity estimation.

**Manifold capacity** ($\alpha = P/N$) refers to the maximum number of object manifolds ($P_{max}$) that can be linearly separated given $N$ features, and characterizes the linearly-decodable object information per feature dimension. MFTMA estimates $\alpha$ through measures of the manifold dimension ($D_M$), manifold radius ($R_M$), and manifold center correlation, which refer to dimensionality, size, and distribution of object manifolds relevant for the linear classification. In our analysis, we combine the manifold dimension and radii together into a single **manifold width** measure, defined as $R_M \cdot \sqrt{D_M}$, which captures the width of the convex hull of a manifold. Manifold width formally links the linear separability of object manifolds with their underlying geometric structure. Specifically, small values of manifold width yield more linearly separable manifold geometry. The correlation between locations of the object manifolds also plays a role in determining manifold capacity, as more correlated manifolds are less separable. For our analysis, center correlation is calculated as the average of the absolute value of the cosine similarity between pairs of object manifold centroids.

In this work, we characterize two types of manifolds:

1) **Class manifolds**: each manifold is defined by the activations evoked by multiple examples drawn from a specific class of visual or auditory stimuli (i.e., object or word identity). The variability within the manifold can come from different exemplars within the class, but may also be influenced by adversarial perturbations, and/or a layer with stochastic activations.

2) **Exemplar manifolds**: each manifold is defined by the activations evoked by multiple instances of a single exemplar (i.e., image or utterance) with the manifold variability reflecting either norm-bounded adversarial perturbations[2] and/or the influence of stochasticity.

Although the class manifold analysis is most tightly coupled to the accuracy of the network on a classification task, analyzing the exemplar manifolds gives insight into how the class manifolds are constructed. Specifically, multiple types of exemplar manifold geometries can lead to the same class manifold capacity and width (Figure 1).

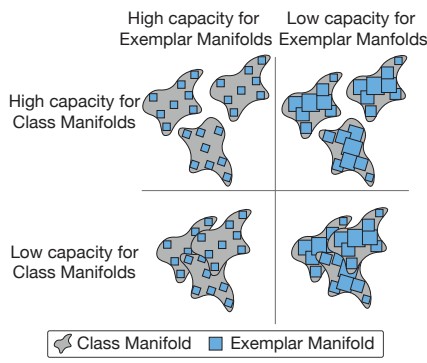

Figure 1: **Insights from class manifolds and exemplar manifolds.** Class manifolds have variability due to differences in exemplars in the class in addition to the stochasticity or adversarial perturbations, while exemplar manifolds only have variability due to stochasticity or adversarial perturbations.

## 3.2 Quantifying object manifold overlap

The MFTMA framework provides a geometric description of exemplar manifolds evoked by adversarial and clean stimuli in the case of multiple distributed object manifolds. However, we also characterize the overlap between adversarial and clean exemplar manifolds generated from the same stimuli (on the grounds that overlap should produce robustness to the adversarial perturbations in question). To do so, we use a notion of object manifold overlap defined as the generalization error of a linear Support Vector Machine (SVM) fit to separate two sets of representations, and report the chance-normalized error rate on the held-out data. A normalized error rate of 1 means that two

---

[2]We replicate many experiments for non-adversarial but random perturbations within the $\epsilon$-sized ball to directly compare model geometry measured on the same stimuli. Results are similar and provided in SM 6.

representations are completely overlapped, and 0 means that they are completely separable and thus not overlapped. We use the `scikit-learn` SVM implementation with a train/test split of 80/20.

### 3.3 Adversarial attacks

For performing adversarial attacks, we use either the single-step fast gradient sign method (FGSM) with a random starting location or multi-step projected gradient descent (PGD). Unless otherwise specified, we use $L_\infty$ norm constrained attacks. All attacks are untargeted and performed on a model-by-model basis. When the goal is to evaluate adversarial accuracy in newly created networks with stochastic internal representations, we use ensemble-PGD, where each step is in the mean direction of k samples of the noisy gradient to ensure a useful signal [36]. For more details, see SM 2.

## 4 Manifold analysis of robustness in ImageNet-trained networks

We first investigate the properties of ImageNet-trained models, aiming to compare the geometric signatures of standard, adversarially trained, and stochastic networks.

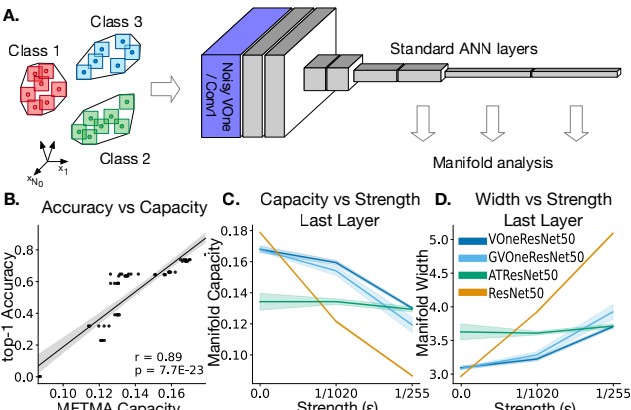

### 4.1 Models and Dataset

We use images sampled from the ImageNet [37] test set. For class manifold analysis, the clean stimulus set consists of 50 classes, with each class containing 50 unique exemplar images for a total of 2500 unique images. For exemplar manifold analysis, 100 unique images are sampled from the ImageNet test set and each is perturbed with FGSM from a random starting location 50 times for 5000 unique images. We analyzed three publicly available ImageNet models, including ResNet50 with standard training, ResNet50 adversarially trained with an $L_\infty = 4/255$ penalty (ATResNet50), and VOneResNet50, a ResNet50-based model with the first conv-relu-maxpool layers replaced by a linear-nonlinear-Poisson model front-end (called the VOneBlock)

Figure 2: **Geometry and capacity of clean and adversarially perturbed class manifolds in ImageNet models** (**A**) Clean or adversarially perturbed stimuli grouped into class manifolds are provided as input to a model, internal representations are extracted, and MFTMA analysis is applied. (**B**) MFTMA capacity from the final model layer reflects accuracy from clean and adversarial images. (**C**) Manifold capacity and (**D**) manifold width for the last model layer is plotted against attack strength, revealing enhanced representational invariance in ATResNet50 (green) and to a lesser degree VOneResNet50 (dark blue) and GVOneResNet50 (light blue), while undefended ResNet50 (orange) is least invariant. Error bars are standard deviation (STD) across 5 random projection (RP) and MFTMA seeds.

with Gabor filters and noise fitted to primate neuronal data. In addition, we analyze a novel variant of VOneResNet50 (GVOneResNet50) with additive Gaussian noise scaled to match the mean over all units in the output of the VOneBlock in response to a set of reference stimuli. GVOneResNet50 performs similarly to the original VOneResNet50 in clean and adversarial conditions (see SM 3.3 for more details). While the first two models are deterministic, VOneResNet50 and GVOneResNet50 have stochastic representations at the output of the VOneBlock. For additional details see SM 3.

### 4.2 MFTMA reveals unique robustness strategies for VOneNet and adversarial training

Our analysis begins by observing the effects of norm-bounded, gradient-based adversarial attacks on the class manifold geometry of representations in ResNet50, ATResNet50, VOneResNet50, and GVOneResNet50. For all experiments, images perturbed with PGD $L_\infty$ constraints of $\epsilon \in [0, 1/1020, 1/255]$ are shown to a model and intermediate representations are extracted, randomly projected to 5000 features [38, 39], and analyzed with MFTMA (Figure 2A). Much like previously

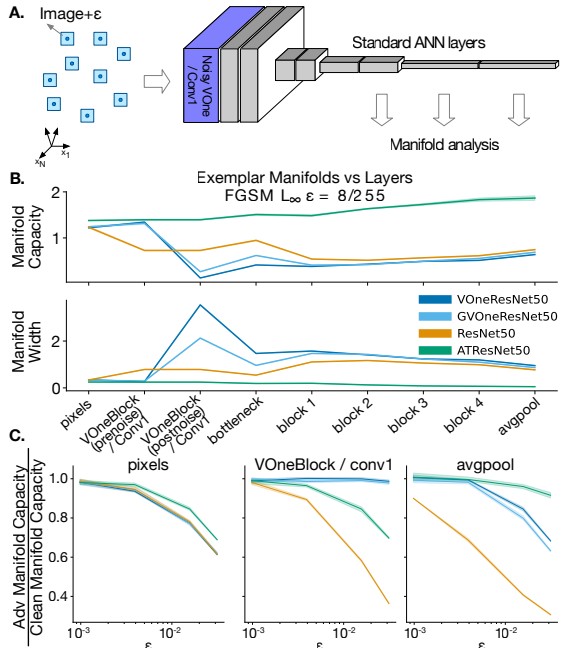

Figure 3: **Geometry and capacity of adversarially perturbed exemplar manifolds in neural networks** (**A**) Each manifold consists of a set of adversarially perturbed examples within an $\epsilon$-sized ball around a single exemplar image. (**B**) Mean capacity and manifold width as a function of layer depth; ATResNet50 maintains the size of $\epsilon$-sized ball manifolds, while undefended ResNet50, VOneResNet50, and GVOneResNet50 expand the size. (**C**) Adversarial representation capacity normalized by clean representation capacity is shown for pixels, first conv layer (the VOneBlock for VOneNets), and the last average pooling layer. VOneResNet50, GVOneResNet50, and ATResNet50 show less of a capacity decrease for low-$\epsilon$ perturbed image manifolds compared to the standard ResNet50. Error bars represent STD across 5 RP and MFTMA seeds in all plots; unnormalized capacity detailed in SM 3.6.

reported with MFTMA [33], models develop separable representations in later layers, with capacity peaking in the final layer before classification (see SM 3.5).

First we focus on this penultimate layer, where class manifold representations have become linearly separable and where capacity thus peaks. For the full layer-wise results, see SM 3.5. We empirically confirm that capacity of the penultimate layer is predictive of the top-1 accuracy across models and attack strengths (Figure 2B), indicating that MFTMA is sensitive to the attacks we are investigating. MFTMA exposes how capacity (Figure 2C) and class manifold width (Figure 2D) vary in the four models as a function of perturbation strength. Adversarial attacks cause the width of class manifolds to grow most rapidly in ResNet50, to a lesser degree in VOneNets, and least of all in ATResNet50.

Next, to investigate how the models represent adversarial perturbations around individual images, we introduce another MFTMA-based approach. Instead of class manifolds, we consider exemplar manifolds consisting of points sampled using FGSM with $L_\infty$ constraints of $\epsilon \in [0, 1/1020, 1/255, 4/255, 8/255]$ from a random starting location in the epsilon-sized ball around 100 exemplar images, with the goal of tracing how these $\epsilon$-sized ball exemplar manifolds develop as they travel through subsequent layers of our networks of interest (Figure 3A).

Restricting our analysis to the layer-wise trajectory of $\epsilon = 8/255$ sized ball exemplar manifolds (Figure 3B), the most salient feature is how distinct the trajectories are for the ATResNet50 and VOneNets. Our results indicate the defense mechanism induced through adversarial training generally stabilizes the width of the $\epsilon$-sized ball around an exemplar as it propagates through the network, effectively mapping small perturbations around the clean image to small regions in later layers of the network. By contrast, the width of the $\epsilon$-sized ball increases in VOneNets and the standard ResNet50. In fact, at the VOneBlock output, the exemplar manifolds become highly entangled, suggesting different mechanisms of robustness for adversarially trained and stochastic models.

How then do stochastic responses improve robustness? Figure 3C demonstrates that when the capacity of the adversarially induced exemplar manifolds is normalized by the capacity for representations of the same unperturbed images (clean exemplar manifolds)[3], VOneNets are far more stable than the standard ResNet50 network. At lower attack strengths where VOneNets' accuracies are minimally degraded, their normalized capacity remains stable, indicating that they represent the perturbed images with approximately the same capacity as clean images. In other words, the adversarial perturbations do not push the representation beyond that for clean images. In SM 3.7, we extend this analysis to a variety of additional networks including two networks trained adversarially with different norms, VOneResNet50 with no stochasticity during training or inference, and ResNet50 with a stochastic activation layer mirroring that in VOneResNet50 (see Dapello, Marques et al. [14]

---

[3]Clean exemplar manifolds for non-stochastic models have the maximum capacity of 2 [20].

for network details.) Our trends hold up in each of these cases, with adversarially trained networks stabilizing the $\epsilon$-sized ball, and stochastic networks expanding it at the point of the stochastic layer, but using similar capacity relative to clean images for small perturbations. In all experiments, we find similar geometric signatures for the Gaussian and Poisson VOneNets. For simplicity in the rest of the paper we focus on stochastic representations with Gaussian noise, but include an analysis of a Poisson noise network trained on CIFAR-10 in SM 5.4.

## 5 Stochastic representations improve robustness in auditory networks

The results from the previous section highlight distinct geometric profiles for adversarially trained models and for models with stochastic responses. To demonstrate that the observed neural population geometry generalizes across modalities, we compare biologically-inspired and adversarially trained auditory models trained to perform a speech recognition task. Our focus here is not on generating an auditory model that is fully defended against adversarial attacks, but rather to test whether the influence of stochastic representations and the geometric trends observed in the previous section generalizes across domains. We thus focus on $L_\infty$ attacks acknowledging that $L_p$ audio attacks that successfully change a network's prediction are often audible to human listeners [40, 41].

### 5.1 Models and dataset

Auditory models are trained to perform the word recognition task in the Word-Speaker-Noise dataset introduced in [9]. Networks learn to distinguish the word present in the middle of a two-second speech clip from 793 word classes. Word class manifolds are constructed from 50 unique words with 50 unique speakers saying the word, drawn from the Wall Street Journal Corpus [42] (2500 unique speech clips). Exemplar manifolds are measured from a random selection of 100 example clips from the class manifold dataset, with 50 samples measured for each clip.

Auditory models contain a biologically-inspired 'cochleagram' representation [43, 44], followed by a ResNet50 architecture. The cochleagram consists of differentiable operations, allowing generation of adversarial examples in the waveform. Unlike VOneNets, and to maintain consistency with previously published auditory models [6, 9], the architecture maintains the conv-relu-maxpoool before the first residual block, on the grounds that the cochleagram models the ear rather than primary auditory cortex. For stochastic models, Gaussian noise with standard deviation $\sigma$ is added after the cochleagram representation (Figure 4A), and we refer to this model as StochCochResNet50. A comparison network without noise ($\sigma = 0$) is similarly trained and evaluated (CochResNet50).

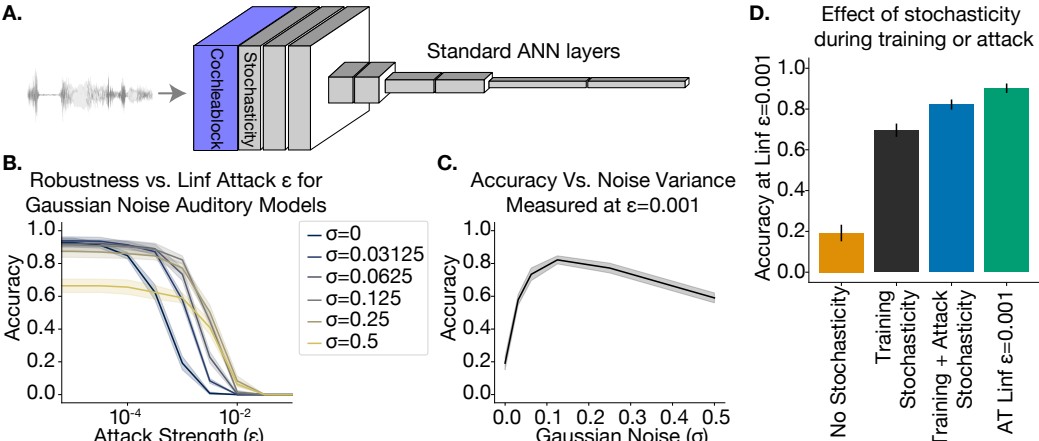

Figure 4: **Robustness of auditory networks trained with stochastic cochlear representations** (**A**) Depiction of auditory network with stochastic cochleagram. (**B**) Performance of auditory networks trained with varying stochasticity levels evaluated on $L_\infty$ adversarial attacks, averaged over 100 randomly chosen test examples of clean speech. Error bars are STD across 5 sets of test stimuli. (**C**) An intermediate level of Gaussian noise yields the best adversarial robustness. (**D**) Stochastic activations during training but not during testing yields improvements in adversarial robustness; maintaining stochasticity during adversarial generation and testing further increases robustness.

A robust model is achieved with adversarial training using $L_\infty$ perturbations with $\epsilon = 10^{-3}$ and maximum of 5 steps for the attack (ATCochResNet50). Training and adversarial robustness details are presented in SM 4.1 and SM 4.4 respectively.

## 5.2 Adversarial robustness in auditory models with stochastic cochleagrams

When setting the level of noise in the VOneBlock, neural responses from Macaque V1 to a specific image set were used to tune the relative amplitude of the stochastic representations. However, this type of neural data is not readily available for all sensory areas. We instead empirically found an optimal level of noise for our auditory models by varying the level of additive Gaussian noise in the stochastic cochleagram. The robustness of these models to $L_\infty$ perturbations is evaluated for different perturbation sizes. To ensure a reliable signal for adversarial attack generation, model gradients are sampled eight times for each PGD iteration. The resulting accuracy for each model is shown in Figure 4B. As noise is increased, the model's robustness to adversarial perturbations increases, but only up to a point, yielding a peak in the robustness curve (Figure 4C). The accuracy for StochCochResNet50 with noise during the training and attack is only slightly worse than ATCochResNet50 (Figure 4D). Much like what was observed in VOneResNet50 [14], including noise during training but not during adversarial evaluation significantly increases adversarial robustness, suggesting that these improvements cannot be trivially explained by the stochastic component masking the gradients during the attack. Instead, the benefits appear to reflect downstream representational changes from training with stochasticity. The additional performance boost when stochasticity is included during inference suggests a secondary defense driven by the stochastic activations during evaluation. These results show that by including stochasticity in a biologically-inspired peripheral model we can improve performance on adversarial examples and by extension learn a more human-like representation.

## 5.3 Neural population geometry of auditory networks

We investigated whether the neural population geometry used by StochCochResNet50 is similar to that observed in the visual domain. As with the ImageNet experiments, clean or perturbed audio is presented to the model and intermediate representations are extracted, randomly projected to 5000 features (if the layer has more than 5000 features), and analyzed with MFTMA.

An analysis of the $\epsilon$-sized ball exemplar manifolds reveals that the geometry of StochCochResNet50 with noise during training but not during evaluation is similar to that of ATCochResNet50, while the StochCochResNet50 with noise during evaluation has a much lower capacity for the adversarial exemplar manifolds at late stages of the network, much like VOneResNet50 (Figure 5A). As in VOneResNet50, when the $\epsilon$-sized ball exemplar manifold capacity is normalized by the clean manifold capacity, the normalized capacity for

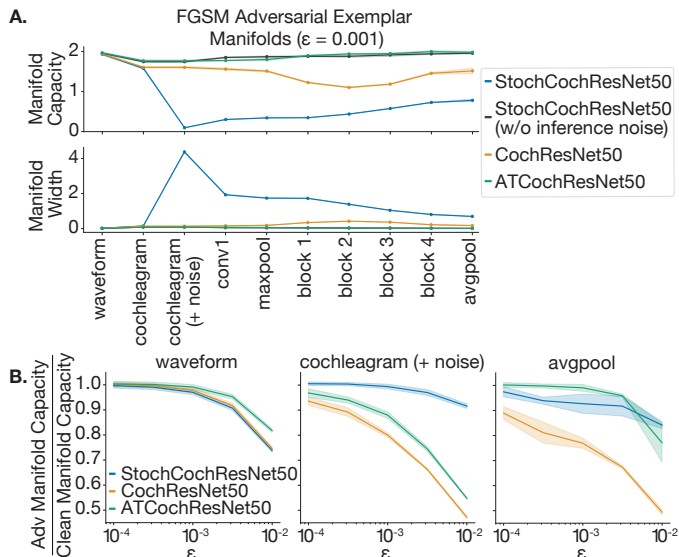

Figure 5: **Adversarial exemplar manifold geometry of auditory models.** (A) Capacity of adversarial exemplar manifolds for auditory models. For networks without stochasticity, the stochastic layer representation is equal to the cochleagram representation. Evaluating the capacity for the adversarial exemplar manifolds with stochasticity present during inference yields significantly lower capacity after the stochastic layer, while the model trained, but not tested, with stochasticity looks similar to the model achieved through adversarial training. Error bars are STD across 5 RP and MFTMA seeds. (B) Adversarial exemplar capacity normalized by the clean manifold capacity as a function of adversarial perturbation size. Unnormalized capacity detailed in SM 4.5. Error bars are STD across 5 RP and MFTMA seeds.

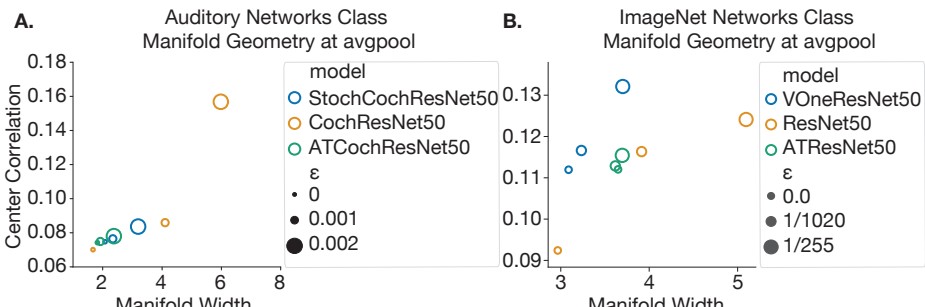

Figure 6: **Similarities between class manifold geometry in Auditory and ImageNet networks.**
Geometry of class manifolds for Auditory (**A**) and ImageNet (**B**) networks when computed from
clean or adversarial stimuli.

StochCochResNet50 is stable with increasing epsilon from the stochastic cochleagram layer onwards
(Figure 5B). This suggests that StochCochResNet50 represents the adversarially perturbed sounds
with the same manifold capacity and width as the clean sounds. By comparison, ATCochResNet50
does not show invariance to the adversarially perturbed stimuli until the avgpool layer.

To examine the key geometric factors associated with an increase in capacity in both the auditory
and visual networks, we plot class manifold width vs. the manifold center correlation, as these two
factors together determine the capacity estimate. Figure 6A and 6B show that in both visual and
auditory networks an increase in size of the adversarial perturbation leads to an increase in manifold
width and center correlation (two key variables leading to the capacity). While this general trend is
present for all networks, the networks that are more adversarially robust have less change across both
metrics as the perturbation size increases. This provides further evidence that the manifold metrics
are a useful way to interpret the internal geometries that lead to adversarial robustness, and points to
the same underlying mechanism in visual and auditory modalities for the classification degradation in
the presence of adversarial vulnerability – specifically that the class manifolds become larger and
more correlated when adversarial perturbations are present.

## 6 Manifold overlap and the opposing effects of noise on robust classification

In the previous sections we investigated how stochastic representations change the size and capacity
of adversarial exemplar manifolds, but we did not directly test whether adversarial examples fall
within the manifolds elicited by stochastic activations. We hypothesize that for small perturbations,
activations elicited from adversarial examples overlap with the exemplar manifold formed by the
stochastic representation of a clean image, effectively eliminating vulnerability to small adversarial
perturbations. Here, we directly investigate whether small adversarial perturbations indeed overlap
with multiple stochastic representations evoked from the same stimulus, and investigate how this
overlap trades off with task performance. To do so, we test whether an SVM can separate the
adversarial exemplar manifolds from the clean exemplar manifolds generated for the same stimulus.
As manifold inseparability as measured by SVM is only one possible description of manifold overlap,
we further detail another analysis using pairwise distance distributions in Section SM 8. Finally, we
analyze the opposing effects of noise on clean and adversarial performance by investigating a CIFAR
model trained with different noise levels.

### 6.1 CIFAR model and datasets

We investigate how the level of noise changes the manifold geometry in a smaller model trained on the
CIFAR-10 dataset [45] with an architecture similar to ResNet18 [2], with the first conv-relu-maxpool
layers replaced by fixed-weight Gabor filters and biological-inspired activation functions adapted
from the Gaussian VOneNet[4]. In SM 5.4, we repeat the experiments with Poisson-like stochastic
activations as well, where the trends are similar.

---

[4]More details about the model architecture and training can be found in SM 5.1

## 6.2 Low perturbation strength adversarial examples overlap with stochastic representations

We measure the overlap between $\epsilon$-sized ball adversarial exemplar manifolds and clean exemplar manifolds at the stochastic representations of the networks (the output of the VOneBlock for the CIFAR and ImageNet models, and the stochastic cochleagram for the auditory model). For all models, the adversarial exemplar manifold is generated by running FGSM $L_\infty$, varying the strength of the adversarial attack. The clean exemplar manifold is generated by measuring multiple stochastic representations from the same natural stimulus. We train an SVM to separate the adversarial exemplar manifold from the clean exemplar manifold (Figure 7). For CIFAR SVM experiments, the stimulus set includes 20 unique image exemplars, each with 1000 samples. For VOneResNet50 and StochCochResnet50 the stimulus set includes 10 unique stimulus exemplars, each with 5000 samples, and the features are downsampled with random projection to 5000 dimensions. In line with our hypothesis, the clean and adversarial manifolds become less overlapped as the adversarial attack strength increases and performance is degraded.

## 6.3 Adversarial robustness requires balancing exemplar and class information

Although increasing the variance of stochastic representations may hide larger adversarial perturbations, high noise levels may also reduce capacity for class manifolds and decrease model accuracy. We hypothesize that the optimal noise level for model accuracy and robustness requires a balance between these two factors (Figure 8A). We empirically test this hypothesis, first by showing that the adversarial and clean exemplar manifolds become less linearly separable with increasing noise variance (Figure 8C). We also evaluate the performance of the CIFAR model with varying noise levels across different adversarial attack strengths (Figure 8D). As the variance of

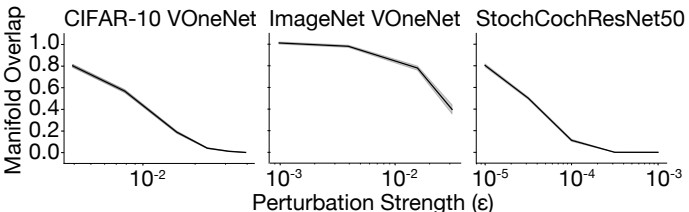

Figure 7: **Adversarial exemplar manifolds measured at stochastic representation overlap with clean exemplar manifolds at smaller attack strengths.** A binary SVM trained to separate adversarial and clean exemplar manifolds extracted from the stochastic representation of each network show that clean and adversarial exemplar manifolds measured from the same stimulus become less linearly separable as the attack strength decreases. Experiments are presented for CIFAR VOneNet (Error bar is STD across 20 images and 6 RP seeds), ImageNet VOneRes-Net50 (Error bar is STD across 10 images and 5 RP seeds) and StochCochResNet50 (Error bar is STD across 10 images and 5 RP seeds).

the stochastic responses increase, the clean performance decreases, while performance under adversarial attack depends non-linearly on the noise variance. Using MFTMA we examine the effect of varying levels of stochasticity on class manifold capacity and related geometric properties. We generated clean exemplar and class manifolds from a stimulus set of 100 unique images, each with 50 noise samples. Figure 8E shows the dependence of manifold capacity on stochasticity level. As the noise level increases, both class and exemplar manifolds become more entangled and capacity drops. In addition, Figure 8F shows that as the noise level increases, the manifold width increases. Thus, both the capacity, which characterizes manifold linear separability, and the geometry demonstrate that increasing noise levels makes manifolds larger, more entangled and less linearly separable. This type of optima is also observed when choosing the level of stochasticity for the auditory networks in Figure 4C, and additional auditory analysis of the noise level is found in section SM 4.6.

## 7 Discussion

Using recently developed techniques to analyze the neural population geometry of a variety of networks on clean and adversarially perturbed stimuli, our work provides new insight into mechanisms underlying adversarial robustness. First, we show key geometrical differences between adversarially trained networks and VOneNets operating with stochastic neural representations. Second, we demonstrate the generality of the usefulness of stochastic representations for defending against adversarial attacks by showing that the effects extend to auditory models. It was not obvious a priori whether stochasticity would have the same benefit on auditory models in part because (unlike vision

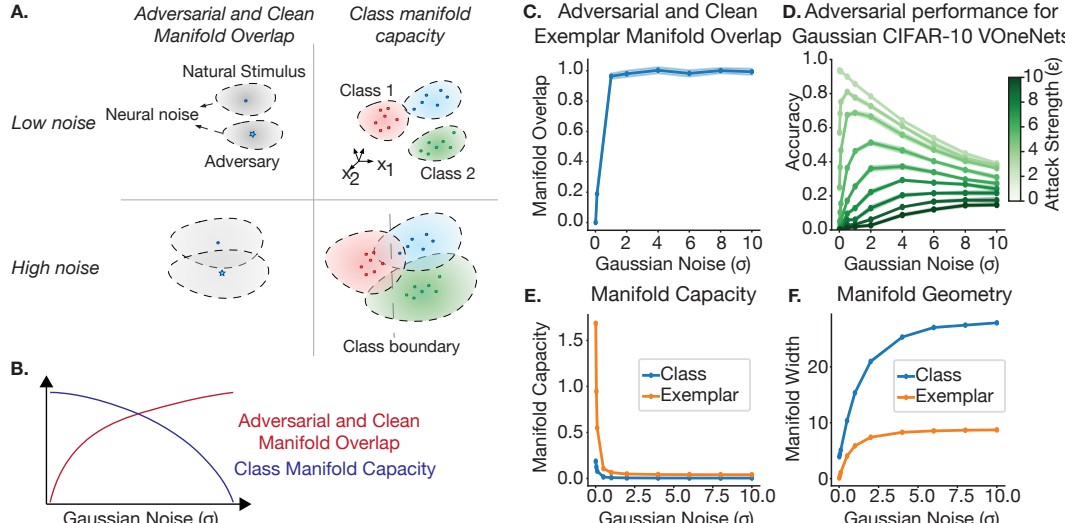

Figure 8: **Stochastic representations induce opposing geometric effects that determine a model's performance** (**A**) Illustration of the stochasticity level's effect on the representations. In the low-noise regime, clean and adversarial representations do not overlap significantly, but class manifolds are well separated. In the high-noise regime, clean and adversarial representations are more overlapped, but the class manifolds also becomes more entangled. (**B**) These two properties trade off as the level of Gaussian noise is varied. (**C**) Binary SVMs are trained to separate adversarial and clean exemplar manifolds representations from same image in CIFAR models, showing that clean and adversarial exemplar manifolds become less linearly separable as stochasticity increases. Error bar is STD from 20 images and 6 random seeds. (**D**) Dependence between model performance and noise level at multiple attack strengths. Error bar is STD from 6 random seeds. (**E**) Capacity for both class and exemplar manifolds capacity decreases as the stochasticity increases. Error bar is STD from 6 random seeds. (**F**) Manifold width increases as the noise level increases. Error bar is STD from 6 random seeds. (All representations are from the stochastic output of the VOneBlock)

models) they are typically trained with additive noise on the input [6, 9] given that additive noise is ubiquitous in real-world auditory signals (because concurrent sounds sum together). Third, we show that stochastic representations of clean and adversarially perturbed stimuli overlap, helping to explain why the stochastic networks generalize better to perturbations below a certain threshold. Fourth, we isolate and quantify the competing effects of stochastic representations on network performance: while stochasticity increases the overlap between clean and adversarial activations, increasing stochasticity also creates more overlap between different class manifolds, making them less separable, and ultimately reducing peak clean network performance.

Here, we focused on linear methods to probe the mechanisms of adversarial robustness. Future work could benefit from more sophisticated methods to measure manifold overlap, and could also extend analysis to more naturally occurring corruptions [46]. Further, while we largely focus on the case of noise injected after fixed biologically inspired filters, more work is needed to explore what types of representations benefit from the addition of noise; for instance Dapello, Marques et al. found there was not as strong of a protective effect with stochasticity added to a standard convolutional filter. We leave the exact role of the neurally plausible filters, as used in both VOneResNet50 and StochCochRenet50, as a promising direction for future work. The presence of noise at all stages of the brain, suggests that if we can resolve when stochasticity is useful, this defense may be extensible to additional network layers for greater gains in robustness.

In theoretical neuroscience, much of the discussion on the role of stochasticity in neural coding has centered around the efficient representation of uncertainties associated with task-relevant variables [47, 48]. Our work adds to this line of research, by using geometry to demonstrate how stochasticity improves the neural population's robustness to adversarial perturbations unseen during training in deep networks. We hope that our work will motivate mechanistic explanations of biologically plausible robust computation in ANNs through the lens of geometry, and further identification of biological constraints that might inspire favorable changes in task-efficient neural representations.

## Acknowledgments and Disclosure of Funding

This work was supported in part by NSF Neuronex 1707398 (S.C.), the Gatsby Charitable Foundation GAT3708 (S.C.), BCS Computational Fellowship (S.C.), Intel Research Grant (S.C., H.L.), a Friends of the McGovern Institute Fellowship (J.J.F.), a DOE CSGF Fellowship (J.J.F.), the PhRMA Foundation Postdoctoral Fellowship in Informatics (T.M), the Semiconductor Research Corporation (SRC) (J.D., J.J.D.), NIH grant R01DC017970 (J.H.M.), the Office of Naval Research grant MURI-114407 (J.J.D.), the Simons Foundation grant SCGB-542965 (J.J.D.), the MIT-IBM Watson AI Lab grant W1771646 (J.D, J.J.D.). All experiments were performed on the MIT BCS OpenMind Computing Cluster.

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
