# Supplementary Material

## SM 1    Manifolds analysis

### SM 1.1    Replica-based Mean Field Theory Manifold Analysis

In this section, we provide a more complete description of Replica-based Mean-Field Theoretic Manifold Analysis (MFTMA). MFTMA refers to the method that was first introduced in [1], and this framework has been used to analyze internal representations of deep networks, ranging from visual [2] to speech [3] and natural language tasks [4]. The shorthand MFTMA was first used in [3], and provided code formed the basis for our analysis[1].

As noted in the main text, object manifolds as used within MFTMA are defined as a set of stimulus evoked representations, grouped by categorical labels. Examples of object manifolds used in this work are population responses to different exemplars in the same object class (a class manifold), or an exemplar manifold created by a variability around a single stimulus (i.e, a single image or an utterance), where variability in the manifold can be from an adversarial perturbation or stochasticity.

Derived using replica theory in statistical physics, the MFTMA framework generalizes the theory of perceptron classification capacity for discrete point patterns [5] to the capacity of object manifolds. Specifically, the MFTMA framework measures the manifold capacity, defined as the maximum number of object manifolds such that the majority of the ensemble of random dichotomy labels for these objects manifolds can be linearly separated. This is a direct generalization of 'shattering' capacity of a perceptron, where the counting unit for the perceptron is the number of objects [2], rather than number of discrete patterns. As the measure of manifold capacity can be empirically evaluated (just as the perceptron capacity can be empirically evaluated), the match between the empirical manifold capacity and the theoretical manifold capacity predicted from the object manifold properties has been shown in many domains with different datasets [1, 2, 3, 4]. The expression for manifold capacity in the MFTMA framework gives rise to new measures for characterizing geometric properties of object manifolds, such that the shattering capacity of object manifolds can be formally expressed in terms of the geometric properties of object manifolds. As the framework formally connects the representational geometric properties and the object manifold's classification capacity, the measures from this framework are particularly useful for gaining a mechanistic account of how information content about objects are embedded in the structure of the internal representations from deep networks. Below we provide additional details of the measures from this framework: manifold capacity and the geometrical properties (such as manifold dimension, radius, width, and manifold center correlation).

### SM 1.2    Metrics in the Manifold Capacity Theory

Given neural or feature representations where $P$ object manifolds are embedded in $N$-dimensional ambient feature (or neural state) space, *load* is defined as $P/N$. Large/small load implies that many/few object manifolds are linearly separable in the feature dimension. Consider a linear classification problem where binary positive and negative labels are assigned randomly to $P$ object manifolds, while all the points within the same manifold share the same label, and the problem is to find a linearly classifying hyperplane for these random manifold dichotomies.

**Manifold capacity**    is defined as the critical load $\alpha_M = P/N$ such that above this value, most dichotomies have a linearly separating solution, and below this value, most of the dichotomies do not have a linearly separating solution. A system with a large manifold capacity has object manifolds that are well separated in the feature space, and a system with a small manifold capacity has object manifolds that are highly entangled (ie, not linearly separable) in the feature space.

Manifold capacity can be estimated using the replica mean field formalism with the framework introduced by [1] and refined in [2]. As mentioned in the main text, $\alpha_M$ is estimated as $\alpha_{MFT}$, or MFTMA manifold capacity, from the statistics of *anchor points*, $\tilde{s}$, a representative point for the

---

[1]https://github.com/schung039/neural_manifolds_replicaMFT
[2]where each object's manifold can include a finite or infinite number of points

points within an object manifold that contributes to a linear classification solution. The general form of the MFTMA manifold capacity has been shown [1, 2] to be:

$$\alpha_{MFT}^{-1} = \left\langle \frac{\left[t_0 + \vec{t} \cdot \tilde{s}(\vec{t})\right]_+^2}{1 + \left\|\tilde{s}(\vec{t})\right\|^2} \right\rangle_{\vec{t}, t_0}$$

where $\langle \ldots \rangle_{\vec{t}, t_0}$ is a mean over random $D$- and 1- dimensional Gaussian vectors $\vec{t}, t_0$ whose components are i.i.d. normally distributed $t_i \sim \mathcal{N}(0, 1)$.

This framework introduces the notion of *anchor points*, $\tilde{s}$, uniquely given by each $\vec{t}$, a coordinate for the manifold's embedded space, and $t_0$, the manifold's center direction, representing the variability introduced by all other object manifolds, in their arbitrary orientations. Formally, $\tilde{s}$ represents a weighted sum of support vectors contributing to the linearly separating hyperplane in KKT (Karush–Kuhn–Tucker) interpretation [1].

Formalized in this way, manifold capacity has many useful interpretations. First, as the manifold capacity is defined as the critical load for a linear classification task, it captures the linear separability of object manifolds. Second, the manifold capacity is defined as the *maximum* number of object manifolds that can be packed in the feature space such that they are linearly separable, it has a meaning of how many object manifolds can be "stored" in a given representation such that they can distinguished by the downstream linear readout. Third, the manifold capacity captures the amount of linearly decodable object information per feature (or neuron) dimension embedded in the distributed representation.

### SM 1.3   Manifold Geometric Measures

The statistics of the anchor points play a key role in estimating a object manifold's effective Manifold Radius $R_M$ and Manifold Dimension $D_M$, as they are defined as:

$$R_{\mathbf{M}} = \sqrt{\left\langle \left\|\tilde{s}(\vec{T})\right\|^2 \right\rangle_{\vec{T}}}$$

$$D_{\mathbf{M}} = \left\langle \left(\vec{t} \cdot \hat{s}(\vec{T})\right)^2 \right\rangle_{\vec{T}}$$

where $\hat{s} = \tilde{s}/\|\tilde{s}\|$ is a unit vector in the direction of $\tilde{s}$, and $\vec{T} = (\vec{t}, t_0)$.

**Manifold Dimension**   measures the effective dimension of the projection of $\vec{t}$ on its unique anchor point $\tilde{s}$, capturing the dimensionality of the regions of the manifolds playing the role of support vectors. In other words, the manifold dimension is the dimensionality of the object manifolds realized by the linearly separating hyperplane. High values of $D_M$ imply that the fraction of the part within the object manifold embedded in the margin hyperplane is high-dimensional, thereby implying that the classification problem is hard.

**Manifold Radius**   measures the average norm of the anchor points, $\tilde{s}(\vec{T})$, capturing the size of the object manifold realized by the linearly separating hyperplane. A small value of $R_M$ implies tightly grouped anchor points.

**Manifold Width**   combines the two measures contributing to the manifold's overall width in the dimensional space, namely, Manifold Radius, $R_M$, and Manifold Dimension, $D_M$. Prior theoretical work has shown that there is a trade-off between $R_M$ and $D_M$ such that as long as $R_M \cdot \sqrt{D_M}$ stays constant, the manifold capacity stays constant [6, 1].

If the object manifold centers are in random locations and orientations, these geometric properties predict the MFTMA manifold capacity [1], by

$$\alpha_{\text{MFT}} \approx \alpha_{\text{Ball}}\left(R_{\text{M}},\ D_{\text{M}}\right) \approx \alpha_{\text{point}}\left(R_{\text{M}} \cdot \sqrt{D_{\text{M}}}\right)$$

where,

$$\alpha_{\text{Ball}}^{-1}(R, D) = \int_{-\infty}^{R\sqrt{D}} Dt_0 \frac{(R\sqrt{D} - t_0)^2}{R^2 + 1}$$

is a capacity of $L_2$ spheres with radius $R$ and dimension $D$ as defined in [6] and

$$\alpha_{\text{Point}}^{-1}(\kappa) = \int_{-\infty}^{\kappa} Dt(t - \kappa)^2$$

is a classification capacity of points given an imposed margin of $\kappa$ as defined in [5, 1].

In real data, the manifolds have various correlations, hence the above formalism has been applied to the data projected into the null spaces of manifold centers, similar to the method proposed by [2]. To characterize the correlation structure in the data, we compute average of absolute values of pairwise cosine correlation between given manifolds' centroids, and provide them alongside other geometric measures here in SM.

### SM 1.4   Capacity: theory vs. simulation

While a good match between empirically observed manifold capacity $(\alpha_M)^3$ and MFTMA predicted manifold capacity $(\alpha_{MFT})$ has been demonstrated in numerous past works [2, 3, 4], here we verify the match between predicted and simulated capacity under adversarial conditions.

In Figure SM1 we show empirical (simulated) capacity vs MFTMA predicted capacity for class manifolds in all analyzed layers of VOneResNet50, ResNet50, and ATResNet50 and under a variety of adversarial perturbation strengths. We observe a tight relationship between MFTMA and simulated capacity, with a mild propensity for MFTMA to overestimate capacity as representations become more separable, consistent with prior observations [2], indicating that MFTMA is also applicable when representations arise from adversarial stimuli.

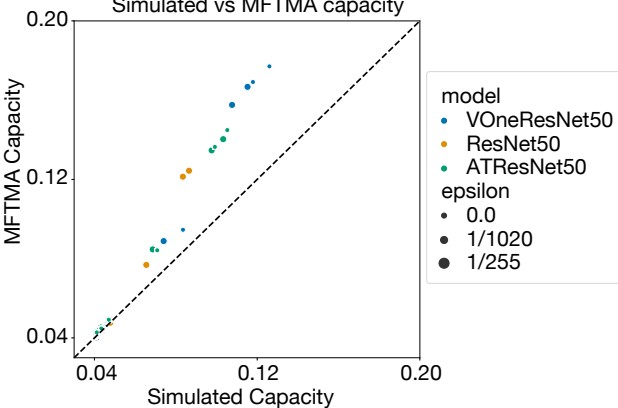

Figure SM1: **MFTMA manifold capacity approximately predicts empirically observed (simulated) manifold capacity under clean and adversarial conditions** for all layers investigated in VOneResNet50 (blue), ResNet50 (orange), and ATResNet50 (green) and across different strengths of adversarial attack, consistent with prior work [2].

---

[3]The empirical capacity is computed using a bisection search on the critical number of feature dimensions required in order to reach roughly 50% chance of having linearly separable solutions given a fixed number of manifolds and their geometries

## SM 2   Adversarial Attacks

For performing white box adversarial attacks, we used the single-step fast gradient sign method (FGSM) [7] or multi-step projected gradient descent (PGD) [8] with an $L_p$ norm constraint. Given an image or waveform $x$, These method uses the gradient of the loss to construct an adversarial image $x_{adv}$ which maximizes the model loss within an $L_p$ bound around $x$. Unless stated otherwise, we use $L_\infty$ attack constraints. Formally, an $L_\infty$ PGD attack iteratively computes $x_{adv}$ as

$$x^{t+1} = \mathsf{Proj}_{x+\mathcal{S}}(x^t + \alpha \, \mathsf{sgn}(\nabla_{x^t} L(\theta, x^t, y)))$$

where $x$ is the original image or audio signal, and the $Proj$ operator ensures the final computed adversarial image $x_{adv}$ is constrained to the space $x + \mathcal{S}$, in this work the $L_\infty$ ball around $x$. FGSM is a special case of PGD, where only a single step is taken.

We used FGSM with a random starting location in the $\epsilon$-sized ball exemplar manifold experiments because we found that when using 64 step PGD for sampling 50 locations around each image, we frequently recovered the same perturbation multiple times, in particular for adversarially trained networks. Therefore, we resorted to FGSM with random starting locations in order to ensure a diversity of sample points in the $\epsilon$-sized ball around our exemplars.

Model specific details of adversarial attacks are provided in each model section below.

## SM 3   ImageNet vision networks

### SM 3.1   Model architecture and training details

With the exception of the Gaussian VOneResNet50, (GVOneResNet50), all of the ImageNet [9] trained models investigated including VOneResNet50, ResNet50, and ATResNet50 were drawn from publicly available sources. ResNet50 pre-trained on ImageNet was taken from `https://pytorch.org/vision/0.8/models.html`. ATResNet50 (adversarially trained on ImageNet with PGD $L_\infty$ $\epsilon = 4/255$) was taken from `https://github.com/MadryLab/robustness` [10]. VOneResNet50 was taken from `https://github.com/dicarlolab/vonenet`. ResNet50 and ATResNet50 share the same architecture, as described in [11]. VOneResNet50 and GVOneResNet50 have the first conv-relu-maxpool of a ResNet50 architecture replaced with a linear-nonlinear model, followed by Poisson-like or Gaussian noise respectively (VOneBlock). More specifically, the VOneBlock consists of a Gabor filter bank tuned to match primate primary visual cortex neuronal data, simple and complex cell non-linearities, and a stochastic layer. The stochastic layer of GVOneResNet50 was composed of zero-mean Gaussian noise with the standard deviation matched to the overall mean across all VOneBlock units in response to a reference stimulus set of natural images. The stochastic layer of VOneResNet was a continuous, second order approximation of Poisson noise as described in Dapello, Marques et al. All components of VOneResNet50 and GVOneResNet50 including the VOneBlock are fully differentiable, and the models are always adversarially attacked end-to-end. For more details on VOneNets, we refer the reader the main text and supplemental materials of [12].

GVOneResNet50 was created using the public `https://github.com/dicarlolab/vonenet` repository. Like existing VOneNets, GVOneResNet50 was trained on ImageNet, with standard preprocessing and data augmentation during training including random resizing and cropping to $224 \times 224$ pixels and random horizontal flipping. For validation, images are center cropped to $224 \times 224$ pixels. Preprocessing was followed by a normalization to render all pixel values between 1 and -1. We used a batch size of 256 images and trained on 2 QuadroRTX6000 GPUs for 70 epochs on the MIT BCS OpenMind computing cluster, for a total training time of approximately 80 hours. We used a step learning rate schedule with 0.1 starting learning rate, divided by 10 every 20 epochs with Stochastic Gradient Descent, a weight decay 0.0001, momentum 0.9, and cross-entropy loss between image labels and model predictions (logits).

Intermediate layers selected for analysis from ResNet50 and ATResNet50 include the pixels, the first conv-relu (Conv1), the maxpool (bottleneck), the output of each residual block (block1, block2, block3, block4), and the average pooling layer (avgpool) prior to the softmax classification layer. For VOneResNet50, and GVOneResNet50 selected layers include pixels, the VOneBlock before the stochastic layer and after the stochastic layer, the $1 \times 1$ convolution following the VOneBlock

(bottleneck), the output of each residual block (block1, block2, block3, block4), and the average pooling layer (avgpool) layer prior to the softmax classification layer. To make layerwise plots well aligned, for ResNet50 and ATResNet50, the Conv1 layer is duplicated at the pre and post noise points in the trajectory, essentially plotting them as if they had a noise layer with 0 variance following Conv1.

## SM 3.2   Adversarial attacks

For ImageNet models (VOneResNet50, GVOneResNet50, ResNet50, and ATResNet50), attacks on class manifold images were performed with untargeted PGD, using 64 steps and a step size of $\epsilon/8$. When evaluating network accuracy to adversarial attacks we average over 8 gradient samples at each step of optimization to ensure useful information from stochastic gradients [13]. We did not average gradients across multiple noise samples for the MFTMA experiments on stochastic ImageNet models, because we are not strictly focused on claims of robust accuracy under a worst case attack. For adversarial $\epsilon$-sized ball exemplar manifolds, FGSM with a random starting location was used, and we verified that all 50 sample points generated were unique images. All adversarial attacks for ImageNet models were performed using the adversarial robustness toolbox [14].

## SM 3.3   Characterizing adversarial robustness of ImageNet models

Here, we characterize the adversarial robustness of the VOneResNet50, GVOneResNet50, ATResNet50, and ResNet50. Of particular interest is GVOneResNet50 in comparison to the original VOneResNet50. Figure SM2 shows the strength accuracy curves of VOneResNet50 compared to GVOneResNet50 for an untargetted $L_\infty$-constrained Projected Gradient Descent (PGD) attack with 64 PGD iterations, a step size of $\epsilon/8$, and eight gradient samples at every step. The plot demonstrates that the GVOneResNet50 is only marginally less robust than the original Poisson-like noise VOneResNet50, indicating that Poisson-like stochasticity is not necessary for improvements in adversarial robustness.

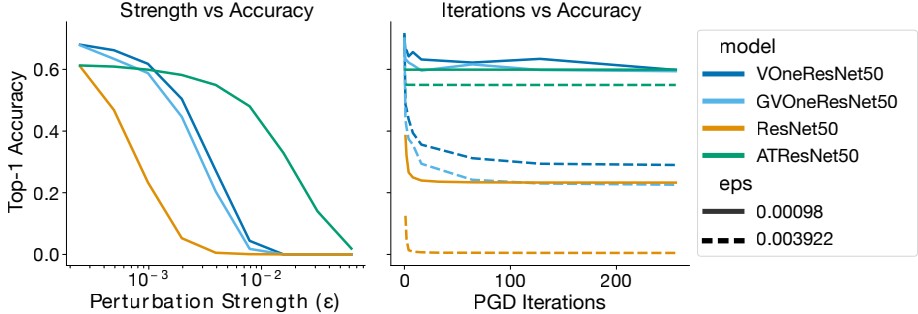

Figure SM2:   **Adversarial robustness of VOneResNet50, GVOneResNet50, ATResNet50 and ResNet50.** Left: the $L_\infty$ PGD perturbation strength ($\epsilon$) vs top-1 accuracy on 5000 ImageNet validation images for VOneResNet50, GVOneResNet50, ATResNet50 and ResNet50. While the original Poisson-like noise VOneResNet50 is slightly more robust than GVOneResNet50, both models are in a comparable range. Right: PGD iterations vs ImageNet top-1 accuracy curve going from zero (random perturbation) to one to many shows that the gradients used in our PGD attack for both models contain useful information for computing adversarial attacks.

Figure SM2 also provides a number of useful sanity checks on the validity of our attacks for these models. For all models, accuracy clearly transitions smoothly from near clean level performance to 0% accuracy as strength increases, indicating that gradients include useful signal for computing optimal image perturbations. Furthermore, increasing the number of gradient iterations from zero (a random starting point) to one and again to many iterations generally increases the effectiveness of the attack, again demonstrating the quality of the gradients for computing adversarial attacks. In general, we did not find multiple random starting points or increasing the number of gradient samples per step beyond eight to have any significant effect on model accuracy.

## SM 3.4   Accuracy vs. capacity

In Figure SM3 we include a more detailed form of Figure 2B, depicting how models and attack strength conditions tested for adversarial accuracy relate to MFTMA capacity.

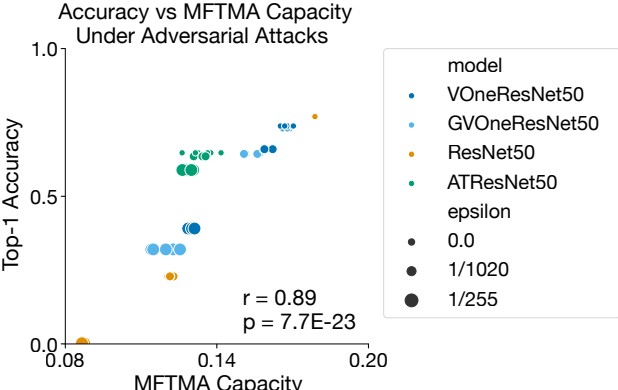

Figure SM3: **MFTMA predicted capacity of class manifolds is well correlated with a model's performance under adversarial attack.** Models include VOneResNet50 (blue), GVOneResNet50 (light blue), undefended ResNet50 (orange), and ATResNet50 (green) for clean images and PGD $L_\infty$ $\epsilon = 1/1020$ and $\epsilon = 1/255$ perturbed images.

## SM 3.5   Full layerwise trajectories of class manifold geometry

Figure SM4 extends the class manifold analysis performed in Figure 2 to show the MFTMA and dimensionality measures from all network stages of VOneResNet50, ResNet50, and ATResNet50. Manifold capacity, $R_M$, $D_M$, center correlation, and the number of principal components needed to capture 90% of the variance were all measured from the class manifold dataset in used in Figure 2. As expected, the class manifold capacity increases in deeper layers of the network and is highest at the final stage of the network (avgpool) for all tested networks.

## SM 3.6   Exemplar manifold capacity for different adversarial strengths

Providing additional context for Figure 3C, Figure SM5 shows the raw manifold capacity (not normalized by the capacity for clean exemplars) of VOneResNet50, GVOneResNet50, ResNet50, and ATResNet50. All networks and all layers considered are above the theoretical lower bound of 0.04 for capacity (given by $2/M$, where M is the number of example points in each manifold.)

When constructing Figure 3C, the values in SM5 were divided by the clean exemplar manifold capacity for each network (ie manifolds measured from unperturbed stimuli and containing variability only from the stochastic activations, if stochasticity was present in the model). Clean exemplar manifold capacity is set to the theoretical upper bound of 2 in all deterministic layers, which is equivalent to treating the manifold as a point. For VOneResNet50, clean exemplar manifold capacity is 0.13214 at the VOneBlock, and 0.90949 at the average pooling layer. For GVOneResNet50, clean exemplar manifold capacity is 0.27230 at the VOneBlock, and 1.01192 at the average pooling layer.

## SM 3.7   Normalized exemplar capacity for additional models

Here, we include several additional ImageNet trained models to confirm the generality of our findings. To look at adversarially robust models beyond the ResNet50 adversarially trained with an $L_\infty$ norm of 4/255 (commonly ATResNet50, here ATResNet50.$L_\infty = 4$) used in the main text, we investigate two additional adversarially trained ResNet50s, one with a stronger $L_\infty$ penalty of 8/255 (ATResNet50.$L_\infty = 8$), and another with an $L_2$ penalty of 3.0 (ATResNet50.$L_2 = 3$)[4]. To isolate the influences of stochasticity compared to the influence of the fixed VOne representation, we also include a ResNet50 with stochasticity added after the initial conv-relu-maxpool block (NoisyResNet50,

---

[4]Additional adversarially trained models were taken from `https://github.com/MadryLab/robustness` [10]

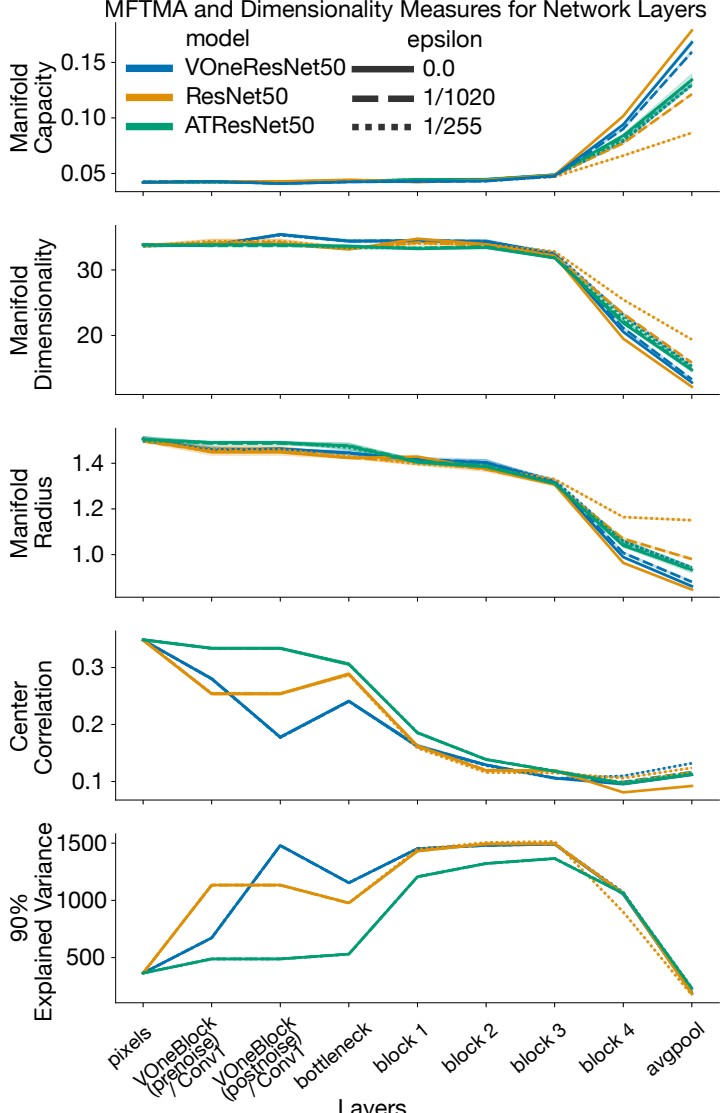

Figure SM4: **Class manifold measures across layers** for VOneResNet50 (blue), ResNet50 (orange), and ATResNet50 (green) across different attack strengths. From top to bottom: mean manifold capacity, manifold dimension, manifold radius, manifold center-center correlation, and the number of principal components need to retain 90% of the total data variance. Error bars represent standard deviation (STD) for 5 RP and MFTMA seeds.

see [12] for full details), and a VOneResNet50 without stochasticity during training or inference (VOneResNet50.NoNoise, see [12] for full details.) These are compared to ResNet50, VOneResNet50 and GVoneResNet50 from the main text. As shown in Figure SM6, the adversarial trained models in green all travel together, the noisy models in blue travel together, and the non-stochastic, non-adversarially trained models in orange also travel together, reflecting distinct geometries due to adversarial training and in models with stochastic activations.

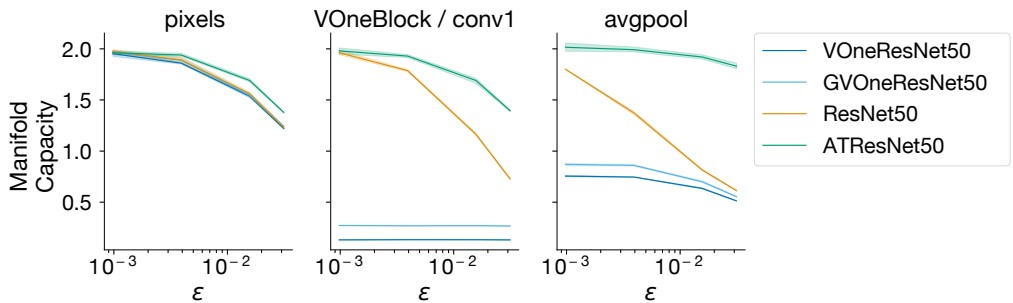

Figure SM5: **Unnormalized adversarial $\epsilon$-sized ball exemplar manifold capacity for pixels, VOneBlock / Conv1, and the final average pooling layer.** Figure 3C without normalization by the clean exemplar manifolds. Error bars are STD for 5 RP and MFTMA seeds.

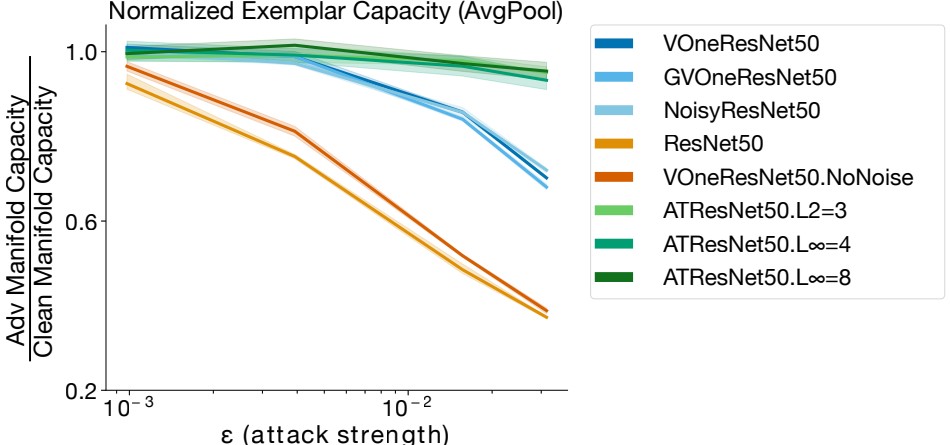

Figure SM6: **Normalized exemplar capacity for additional models.** relative exemplar manifold capacity (adversarial exemplar manifold capacity normalized by clean exemplar manifold capacity) is show for the average pool layer of adversarially trained models in green including ATResNet50.$L_2 = 3$ (light green), ATResNet50.$L_\infty = 4$ (green), and ATResNet50.$L_\infty = 4$ (dark green), stochastic models in blue VOneResNet50 (blue), GVOneResNet50 (light blue), and NoisyResNet50 (lighter blue), and non-stochastic, non-adversarially trained models in orange including ResNet50 (orange) and VOneResNet50.NoNoise (dark orange) show that stochasticity and adversarial training both lead to unique signatures of robustness across a range of networks.

## SM 4 Auditory networks

### SM 4.1 Model architecture and training details

All auditory models (CochResNet50, ATCochResNet50, and StochCochResNet50) included all components of the standard ResNet50 architecture [11]. Rather than the first convolutional layer acting on an image, the first convolution is applied to the generated cochleagram representation.

The 'cochleagram' representation is similar to a spectrogram, but with frequency resolution and compression tuned to approximate the human ear. Cochlear filters were constructed using the pycochleagram library (https://github.com/mcdermottLab/pycochleagram). The cochlear model consists of a filterbank of 211 bandpass filters with frequency response as the positive portion of a cosine function, spaced on an equivalent rectangular bandwidth (ERB) scale with low limit of 50Hz and high limit of 10kHz, including lowpass and highpass filters [15, 16]. Audio input to the networks was two seconds long sampled at 20kHz (40,000 samples). Passing audio through these filters results in audio subbands, and the envelope of the each is computed via the analytic amplitude of the Hilbert Transform. Envelopes are downsampled to 200Hz and passed through a compressive nonlinearity ($x^{0.3}$). The output of this yields a cochleagram representation of shape (211, 390), which served as the input to the standard ResNet50 architecture layers. The cochleagram operations were implemented in PyTorch, and all components of the cochleagram generation are differentiable, allowing adversarial examples to be generated directly on the waveform in an end-to-end manner.

In StochCochResNet50, a layer of additive Gaussian noise with a mean of zero was applied to the cochleagram representation before being passed to the first convolutional layer of the ResNet50 architecture. For auditory model analyses, we investigated the audio input to the network (waveform), the output of the cochlear model (cochleagram), the additive Gaussian noise stochastic layer (cochleagram + noise), the first conv-relu (conv1) of ResNet50, the first maxpool (maxpool) of ResNet50, the output of each residual block (block1, block2, block3, block4), and the average pooling layer (avgpool) that occurs before the logits of the ResNet50 architecture. To align layerwise plots, the "cochleagram (+ noise)" layer is a duplication of the "cochleagram" layer in detemistic models CochResNet50 and ATCochResNet50, while it is the "cochleagram + noise" for StochCochResNet50.

Auditory networks were trained in PyTorch 1.5.0 on the word recognition task from the Word-Speaker-Noise dataset introduced in [17]. Audio sampling rate was 20kHz. Samples from the audioset dataset served as additive noise in the waveform, and were combined with the speech clips at uniformly selected signal-to-noise ratios of -10dB to 10dB. A random 2 second crop of the speech signal was extracted, always ensuring that the labeled word overlapped with the 1 second boundary of the signal, and a random 2 second crop of the audioset background was added to the speech signal. The combined audio was mean subtracted and normalized to a root-mean-square level of 0.1 before being processed by the cochlear model. Adversarial performance curves were evaluated on held out speech clips, randomly cropped and normalized the same as during training (but excluding the audioset augmentation).

Models were trained with a batch size of 256 on 8 Nvidia Tesla-V100 GPUs on the MIT BCS OpenMind computing cluster. Stochastic models and the standard network took approximately 18 hours to train, while the adversarially trained network took approximately 114 hours. Each model was trained with 150 epochs of the speech data (corresponding to 42 epochs of the audioset clips). Learning rate started at 0.1 and was divided by 10 after every 50 epochs, using the pytorch SGD optimizer with momentum 0.9, weight decay 0.0001, and a cross entropy loss between the word labels and model predictions. All training was performed with a modified version of the robustness library [10] with additions to handle auditory training.

The adversarial training parameters for ATCochResNet50 consisted of a $L_\infty$-norm bound of 0.001 on the waveform. Five attack steps were applied for the PGD attack during training, starting at a random location with a step size of 0.001/2.

### SM 4.2 Adversarial attacks

The adversarial robustness of audio models (StochCochResNet50, CochResNet50, ATCochResNet50) to untargeted $L_\infty$ attacks (Figure 4 and Figure SM7) was evaluated using 32 PGD steps with step size of $\epsilon/5$. For the stochastic models (StochCochResNet50) model gradients were sampled eight

times for each PGD iteration and averaged to obtain the step direction. Analysis of the adversarial robustness with this ensemble method was conducted with [14].

For adversarial $\epsilon$-sized ball exemplar manifolds in Figure 5, FGSM with a random starting location was used to measure 50 samples for each audio exemplar. The attacks were untargeted $L_\infty$ attacks with step size of $2\epsilon$. For analyses of adversarial class manifolds in Figure 6A, a single adversarial example was generated for each audio sample in the class using 32 steps of PGD. An untargeted $L_\infty$ attack with a random starting location and step size of $\epsilon/5$ was used for constructing these adversarial class manifolds. Similar to VOneResNet50 and CIFAR-VOneNet, we did not average over multiple gradient samples when constructing the manifolds for the stochastic audio models, as for these experiments we were focused on generating samples from the adversarial manifolds rather than on evaluating the models defenses. Adversarial examples for both types of manifold experiments were obtained using [10].

### SM 4.3    Choice of Gaussian noise level for StochCochResNet50

The evaluation shown in Figure 4C showed that a Gaussian noise level of $\sigma = 0.125$ yielded maximum adversarial robustness at $\epsilon = 0.001$ to untargeted $L_\infty$ attacks. This level of Gaussian noise also yields best performance when averaged across all tested $\epsilon$ values (Figure SM7). Thus, Gaussian noise level of $\sigma = 0.125$ was used for the presented experiments that made comparisons between StochCochResNet50, CochResNet50 and ATCochResNet50.

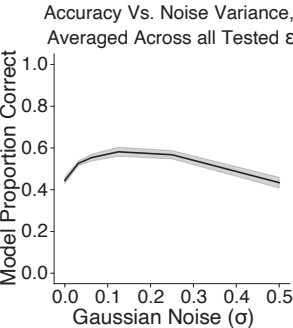

Figure SM7: **Adversarial performance of StochCochResNet50 models** Analysis of model performance averaged across all tested $L_\infty$ attack $\epsilon$ values compared to the level of Gaussian noise in the model, evaluated over 100 randomly chosen speech examples. Model performance peaks at $\sigma = 0.125$. Error bars are STD across 5 sets of test stimuli.

We further quantified the the Signal-to-Noise-Ratio (SNR) of the stochastic cochleagrams for the selected Gaussian noise levels by (1) calculating the mean cochleagram across 20,000 examples from the training data (2) taking the mean across time and frequency (3) dividing this value by the standard deviation used for the Gaussian noise at the stochastic layer. These SNR values are reported in SM Table 1. Note that the best model shown in bold is close to the SNR ratio of 1 that was chosen for GVOneNet. A histogram of the averages for the cochlear channels and the ordered average for each cochlear channel is shown in Figure SM8 to demonstrate the distribution of average channel activations.

| Gaussian ($\sigma$) | StochCoch SNR |
|---|---|
| 0 | inf |
| 0.03125 | 4.4433 |
| 0.0625 | 2.2217 |
| **0.125** | **1.1108** |
| 0.25 | 0.5554 |
| 0.5 | 0.2777 |

Table 1: SNR of the cochleagram + noise layer of the StochCochResNet50 architecture when changing the standard deviation ($\sigma$) of the additive Gaussian noise. Model with best performance across adversarial attacks is shown in bold.

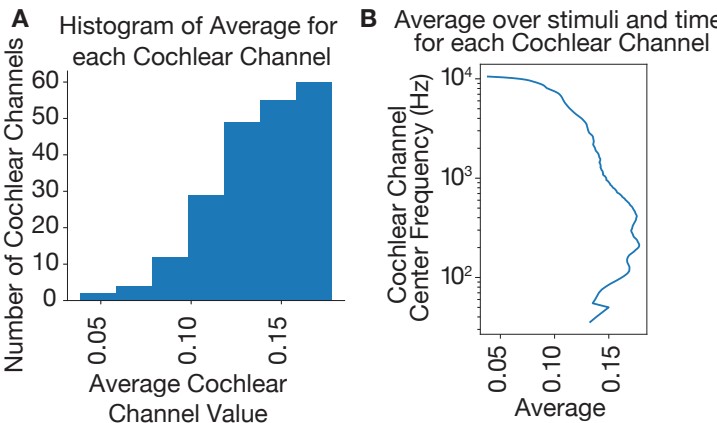

Figure SM8: **Average cochlear channel values**. (A) Histogram demonstrating the distribution of the average value across time in each cochleagram channel. (B) The time average of each cochleagram channel, ordered by frequency.

## SM 4.4   Characterizing adversarial robustness of StochCochResNet50

We further validated the adversarial robustness of the StochCochResNet50 with $\sigma = 0.125$ to $L_\infty$ attacks by sweeping through different numbers of PGD iterations and step sizes (Figure SM9). The StochCochResNet50 with stochasticity during inference remained more robust than the StochCochResNet50 without stochasticity during inference, and both StochCochResNet50 evaluations were more robust than the CochResNet50.

As with our sanity checks on VOneResNet50 and GVOneResNet50, Figure SM9 results demonstrate that going from random perturbations (zero iterations) to one PGD iteration increases the effect of the attack, and again from one to many iterations the effect of the attack is increased, indicating that the gradients are not broken and indeed contain information for computing adversarial perturbations.

We further tested the adversarial robustness of StochCochResNet50 to $L_2$ perturbations. Similar to the $L_\infty$ results in Figure 3D, the StochCochResNet50 with stochasticity during inference was more robust than the StochCochResNet50 without stochasticity during inference, however both were more robust to $L_2$ perturbations than the the CochResNet50 with no adversarial defenses. The network trained with $L_\infty$ adversarial perturbations (ATCochResNet50) was more robust than all networks. We

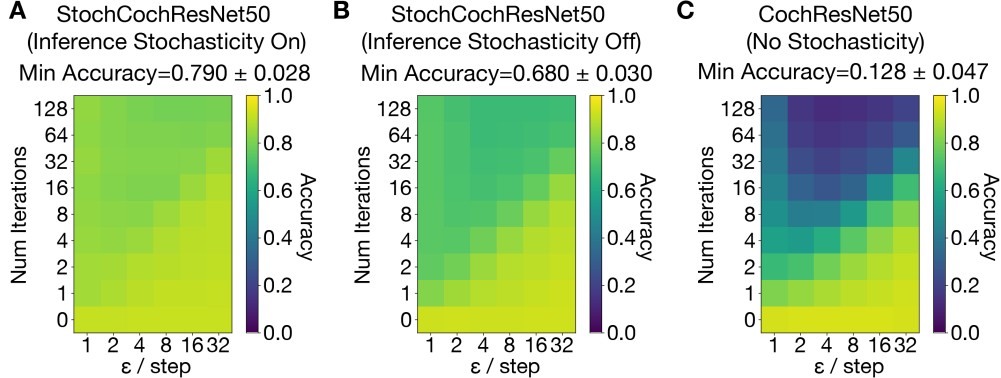

Figure SM9: $L_\infty$ **adversarial evaluation of Auditory Networks**. StochCochResNet50 is adversarially attacked with (A) and without (B) including stochasticity during adversarial generation and inference. CochResNet50 (C) is similarly adversarially attacked. The number of iterations and the size of the attack step is varied for each model, with $L_\infty$ attack strength of $\epsilon = 0.001$. For StochCochResNet50 with inference, the gradients are averaged over 8 instantiations of the model. The worst accuracy across all attack possibilities is reported, and the STD is computed across 5 sets of test stimuli.

performed a similar analysis to Figure SM9, using $L_2$ attacks. Once again, for StochCochResNet50 with inference stochasticity, we see that by increasing the number of iterations the performance decreases, indicating that the gradients for the stochastic networks maintain information to compute adversarial perturbations. For all evaluations of StochCochResNet50 with inference stochasticity, we found that an ensemble size greater than eight no longer improves effectiveness of the adversarial attacks.

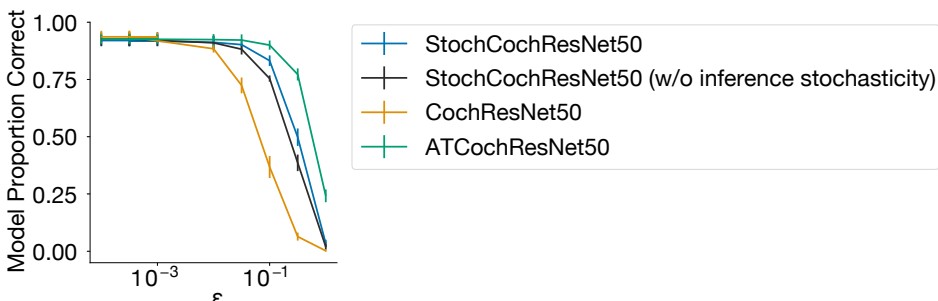

Figure SM10: $L_2$ **evaluation of Auditory Networks for different $\epsilon$ sizes**. Adversarial performance of auditory networks on $L_2$ adversaries for various $\epsilon$ sizes. Step size was set to $\frac{\epsilon}{8}$ and generated from 32 PGD iterations. For StochCochResNet50 with inference, gradients for each iteration were averaged over 8 instantiations of the model.

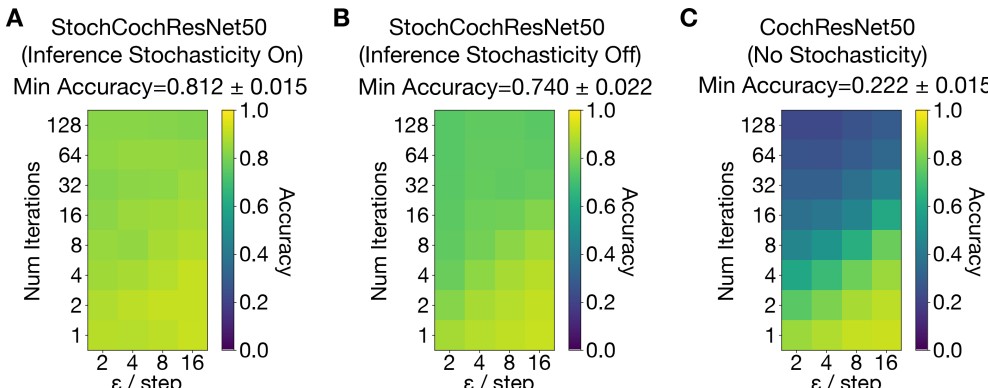

Figure SM11: $L_2$ **adversarial evaluation of Auditory Networks**. StochCochResNet50 is adversarially attacked with (A) and without (B) including stochasticity during adversarial generation and inference. CochResNet50 (C) is similarly adversarially attacked. The number of PGD iterations and the size of the attack step is varied for each model, with $L_2$ attack strength of $\epsilon = 0.1$. For StochCochResNet50 with inference, the gradients are averaged over 8 instantiations of the model. The worst accuracy across all attack possibilities is reported, and the STD is computed across 5 sets of test stimuli.

## SM 4.5   Auditory Networks: Capacity for different adversarial strengths, unnormalized

Figure SM12 shows the raw manifold capacity (not normalized by the capacity for clean exemplars) of StochCochResNet50, CochResNet50, and ATCochResNet50. All networks and all layers considered are above the theoretical lower bound of capacity ($2/M$, where $M$ is the number of points in each manifold, here $M = 50$.) Clean exemplar manifold capacity used for normalization in Figure 5B is 2 for deterministic layers, 0.09690 at the stochastic cochleagram layer of StochCochResNet50, and 0.80927 in the StochCochResNet50 average pooling layer. As in the VOneNet analysis, clean exemplar manifold capacity is set to the theoretical upper bound of 2 in all deterministic layers.

## SM 4.6   Auditory Networks: Opposing factors of stochasticity level

We investigated whether the tradeoff seen in Figure 8 for the CIFAR network was also present for auditory networks. We examined manifolds created from samples within the $L_\infty$ $\epsilon = 0.001$

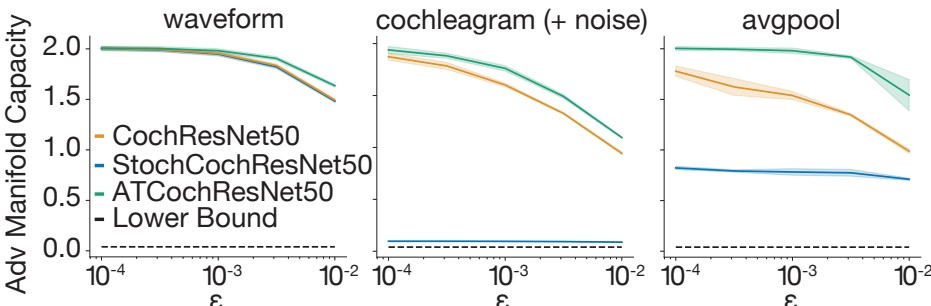

Figure SM12: **Auditory networks adversarial exemplar manifold capacity, unnormalized** Analysis of auditory networks unnormalized $\epsilon$-sized ball exemplar manifold capacity for waveform, cochleagram (+ noise), and the final average pooling layer. Dashed line represents the lower bound for capacity. Error bars are STD across 5 RP and MFTMA seeds.

ball, where we saw high variability in performance across networks with different $\sigma$. To measure the overlap of the clean exemplar and adversarial exemplar manifolds, we test whether an SVM can separate the exemplar noise manifolds from the adversarial perturbation exemplar manifolds generated from the same sound clip (Figure SM13A). Activations for 5000 samples of each exemplar are measured at the "cochleagram + noise" representation and downsampled to 2048 dimensions. Similar to the CIFAR results, as the noise increases the SVM error rate increases as the clean exemplar and adversarial exemplar manifolds become more entangled. The tradeoff with manifold capacity is revealed in the Class (Figure SM13B) and exemplar (Figure SM13C) manifold analyses. We analyze the class and adversarial exemplar manifolds from the stochastic layer and from the avgpool layer, constructing the manifolds as described in SM 4.2. As the stochasticity in the network increases, the capacity decreases, until the noise outweighs the signal and the network cannot perform the task.

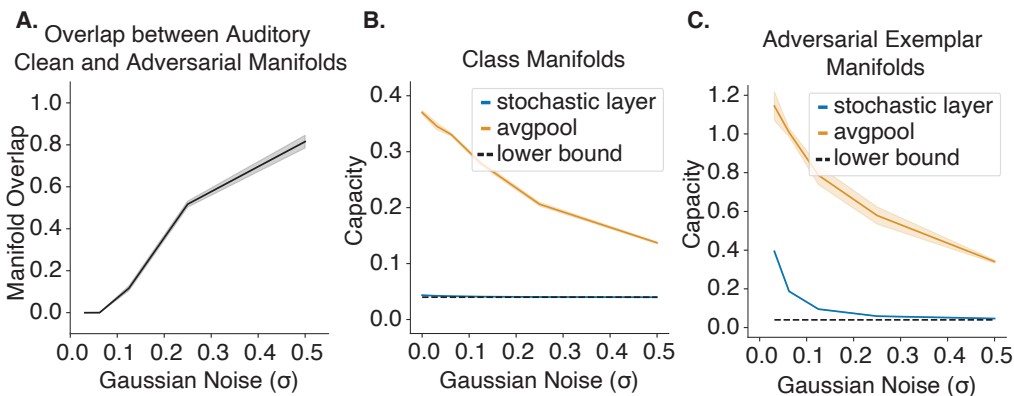

Figure SM13: **Auditory models demonstrate that stochastic representations induce opposing geometrical properties that determine a model's adversarial performance**. (**A**) SVM analysis of the overlap between the clean exemplar and adversarial exemplar manifolds measured at the stochastic "cochleagram + noise" layer of StochCochResNet50 trained and evaluated with varying levels of Gaussian noise. Error bars are STD across 10 sounds and 5 random seeds. (**B**) Class manifold capacity and (**C**) adversarial exemplar manifold capacity for the stochastic layer and the average pooling layer of a StochCochResNet50 trained and evaluated with varying levels of Gaussian noise. Error bars are STD across 4 random seeds.

Figure SM14 shows the MFTMA measured manifold radius, manifold dimension and center correlation for clean class manifolds and for adversarial exemplar manifolds across different Gaussian noise levels in StochCochResNet50 networks, corresponding to the capacity measures in Figure SM13B,C. Manifold radius and dimension increase as the noise increases for both types of manifolds, corresponding to an increased manifold width. The manifold center correlations decrease as noise increases for early layers of the network, but the manifolds become less correlated at late stages of the network with increasing noise.

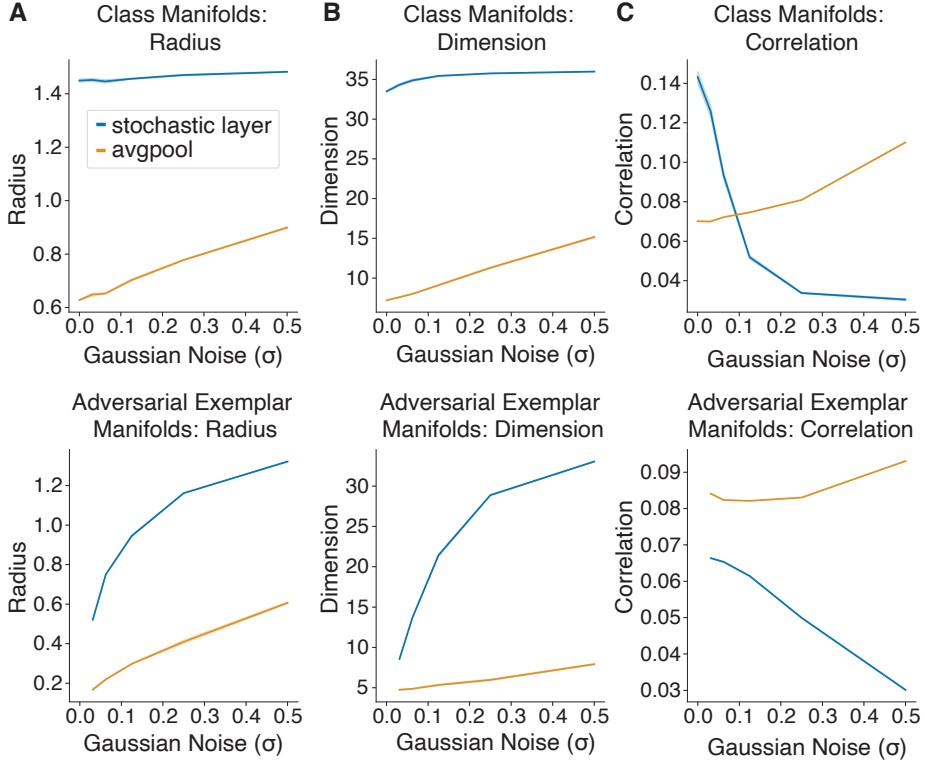

Figure SM14: **Additional geometric properties of StochCochResNet50 when varying level of Gaussian noise (A)** Manifold radius, **(B)** manifold dimension, **(C)**, manifold center correlation of the Gaussian noise model with respect to the level of noise. Similar to the CIFAR results, as the level of noise increases, manifold radius and dimension also increases, while center correlation decreases at the early layer (after noise) and increases at the late layer (avgpool). Error bar is STD from 4 random seeds.

# SM 5 CIFAR10 vision networks

## SM 5.1 Model architecture and training details

| Layer Name | Description |
|---|---|
| pixels | Input $32 \times 32$ color image |
| conv1 (prenoise) | VOneBlock 32 simple channels and 32 complex channels of shape $9 \times 9$, stride 1 |
| conv1 (postnoise) | Gaussian noise layer |
| block 1 | `conv2_x` of ResNet18 |
| block 2 | `conv3_x` of ResNet18 |
| block 3 | `conv4_x` of ResNet18 |
| block 4 | `conv5_x` of ResNet18 |
| avgpool | Average pool layer |

Table 2: CIFAR-VOneNet Architecture

The CIFAR-VOneNet architecture is detailed in Table 2. For the VOneBlock, the architecture specification is based on the implementation of the VOneBlock in the VOnenet model from [12]. There are several modifications from the original VOneBlock. First, the parameters for the Gabor filters including the kernel size and the spatial frequency are scaled down to 50% of their ImageNet size to better fit the small size of the CIFAR-10 dataset. Further, we use 32 simple and 32 complex channels instead of 256 simple and 256 complex channels. Finally, Gaussian noise, instead of Poisson

noise, is added after the simple and complex cell non-linearities of the VOneBlock, for simplicity of tuning stochasticity levels. We note that the modification of the VOneBlock to fit CIFAR-10 inputs means that the original tuning to fit primate primary visual cortex data is disrupted, but we believe the general observations from our experiments on this version hold. We use `PyTorch 1.5` to train the models. The models are trained with a `SGD` optimizer with 120 epochs and an initial learning rate of 0.1. The learning rate is multiplied with a $\gamma$ factor of 0.1 every 40 epochs. The batch size is 128 and the weight decay is 0.0005. The models take approximately 3 hours to train with a NVIDIA Titan X GPU. All models were trained on the MIT BCS OpenMind computing cluster.

In Figure 8, we varied the stochasticity level by scaling the standard deviation of the Gaussian noise. Table 3 provides the corresponding signal-to-noise (SNR) ratio to give more context on the magnitude of the stochasticity with respect to the strength of the feature activations. The signal strength is measured by taking the average activation at conv1 (prenoise) of 10000 images in the CIFAR test set. The signal-to-noise is calculated by dividing this average activation, which is 0.4215, by the Gaussian noise standard deviation. Note that the feature activation is quite sparse after the ReLU non-linearity activation at the end of the VOneBlock, so the mean feature activation may be smaller than the typical positive activation.

| Gaussian $\sigma$ | Activation SNR |
|---|---|
| 0.01 | 42.152 |
| 0.1 | 4.215 |
| 1.0 | 0.422 |
| 2.0 | 0.211 |
| 4.0 | 0.105 |
| 8.0 | 0.053 |
| 10.0 | 0.026 |

Table 3: Corresponding Signal to Noise Ratio with the standard deviation of the Gaussian noise

## SM 5.2   Adversarial attacks

To evaluate the adversarial performance, adversarial images were generated by untargeted PGD with random starting locations within the $\epsilon$-ball. Specifically, the number of iterations was 64 and the step size was $\epsilon/32$. The model gradients used for each PGD step was the average gradient direction of 64 sample gradients to obtain reliable signal.

To generate the adversarial $\epsilon$-sized exemplar manifolds, untargeted FGSM with random starting locations within the $\epsilon$-ball was used. The number of iterations was 1 and the step size was $\epsilon$. The gradient used in FGSM was only sampled once, because in this experiment we focus on generating diverse samples for the adversarial manifolds rather than evaluating the model's adversarial robustness. All adversarial attacks for CIFAR models were performed using the adversarial robustness toolbox [14].

## SM 5.3   Additional geometric properties for CIFAR-VOneNets

In this section, we provide the additional MFTMA measurements including manifold dimensions, manifold radii, manifold center-center correlations, as well as the number of principal components needed to capture 90% of the total representation variance for selected experiments detailing manifold capacity and width in Figure 8 of the main text.

Figure SM15 shows additional geometric properties (manifold radius, dimension and center correlation) for class manifolds and clean exemplar manifolds across different Gaussian noise levels in the CIFAR-VOneNet networks, measured at the stochastic conv1 (postnoise) layer.

## SM 5.4   Replication of CIFAR-VOneNet model with Poisson noise

We replicate the overlap analysis presented in Figure 8 using a CIFAR VOneNet with Poisson noise rather than Gaussian noise (SM16). Similar to the Gaussian noise model, in the Poisson noise model, the noise is injected after the simple and complex cell non-linearities in the VOneBlock. We use the Poisson noise implementation from [12], in which the Poisson noise is approximated by adding a

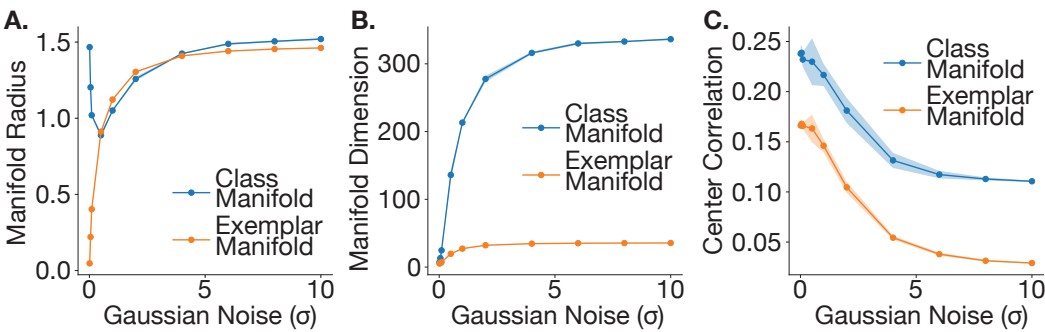

Figure SM15: **Additional geometric properties of CIFAR-VOneNet Gaussian noise model (A)** Manifold radius, **(B)** manifold dimension, **(C)** manifold center correlation of Gaussian noise model with respect to the level of noise, measured at the stochastic layer. As the level of noise increases, manifold radius and dimension also increases, while center correlation decreases. Error bar is STD from 6 RP and MFTMA seeds.

Gaussian noise with mean zero and variance equal to the feature activation. With this formulation of Poisson noise, the variance-to-mean ratio is always 1. To investigate the effect of modifying the level of Poisson noise, we introduce the parameter noise ratio $r$, which represents the variance-to-mean ratio of the injected noise. For example, for the Poisson noise model with $r = 2$, the added noise is Gaussian with mean zero and variance equal to twice the feature activation.

Similar to the Gaussian noise model, as the level of stochasticity increases, the stochastic exemplar adversarial and clean manifolds also become more overlapped (Figure SM16A). Figure SM16B shows the adversarial performance of the Poisson noise model with varying noise levels at multiple different attack strengths. Note that the model is trained and tested with the same noise levels. Compared to the Gaussian noise models, the performance of the Poisson noise model is less dependent on the noise level. Across various attack strengths, adversarial performance increases at low noise ratios and then plateaus, not degraded as in the case of Gaussian noise model, at high noise ratios. This phenomenon can be related to the property that Poisson noise has variance relative to individual unit activation, while Gaussian noise has an absolute variance for all units. Figure SM16C and Figure SM16D show MFTMA analysis for class manifolds and exemplar manifolds for the Poisson noise model with no adversarial perturbations, demonstrating that as the level of stochasticity increases, both the class and exemplar noise manifolds become more linearly entangled. Manifold radius, dimension, and center correlation are similarly shown in Figure SM17. Similar to Gaussian Noise, we see a general tradeoff between the overlapping adversarial and noise manifolds and the manifold capacity. However, compared to Gaussian noise, Poisson noise introduces different variances across individual units within the neural population, therefore introducing distinct population geometries. The nature of how different types of noise statistics affect robust perception through the lens of neural population geometry is an important direction for future studies.

## SM 6    Replication of adversarial exemplar manifold results with random perturbations

Our exemplar manifold results detailed in the main text focused on investigating the geometry of activations measured from manifolds created by FGSM adversarial examples. One possible confound when defining manifolds based on adversarial examples is that different stimuli are used for each network (for instance, there may be a bias when generating the adversarial exemplar manifolds, resulting in some models having more similar examples within each exemplar manifold). As a control for this, we ran the the same exemplar manifold analyses using random points measured on the shell of the $L_\infty$ ball, presenting the same random points to each network. The main results held across these analyses, as documented in the following section. This highlights that the differences in geometry of the networks are measurable with generic stimuli and not a bi-product of the specific set of adversarial examples or attack methods used for the analysis.

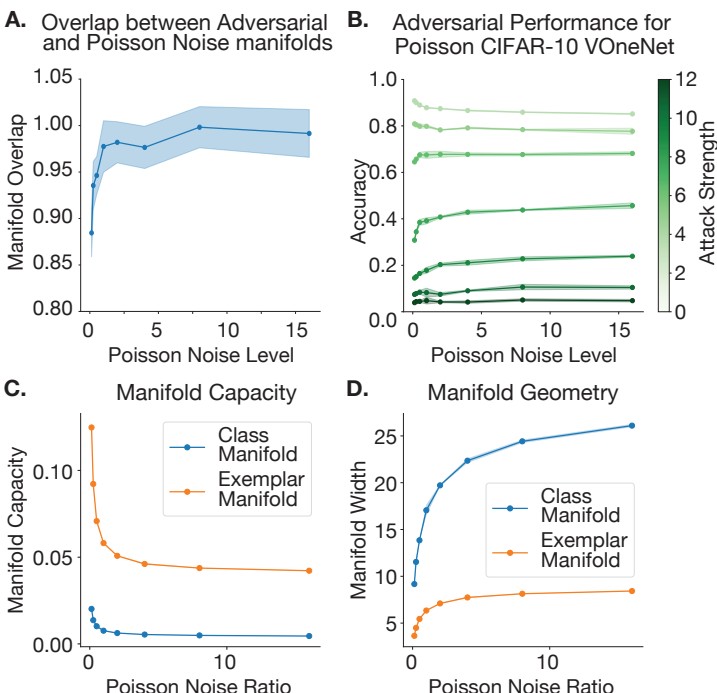

Figure SM16: **Poisson CIFAR-10 networks show opposing geometric effects of stochasticity** Replication of Figure 8 with Poisson CIFAR-10 networks. **(A)** A binary SVM is used to classify Poisson-like noise vs adversarial clouds from the same image, showing they become less linearly separable as the noise level increases, across multiple network layers. Error bar is STD across 20 images and 6 random seeds. **(B)** Adversarial performance across different attack strengths demonstrate the similar trade-off between clean accuracy and adversarial robustness. Error bar is STD across 6 random seeds. **(C)** Manifold Capacity decreases as the level of stochasticity increases. Error bar is STD across 6 random seeds. **(D)** Manifold Width increases as the level of stochasticity increases. Error bar is STD across 6 random seeds.

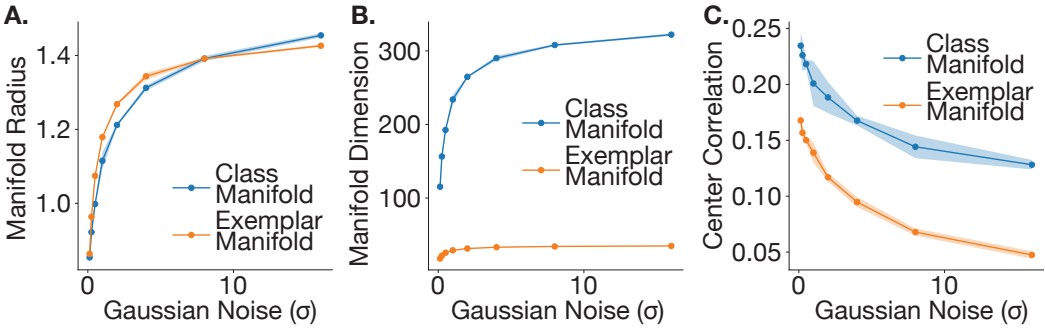

Figure SM17: **Additional geometric properties of Poisson-like noise model (A)** Manifold radius, **(B)** manifold dimension and **(C)** manifold center correlation of Poisson-like noise model with respect to the level of stochasticity. Manifold radius and dimension increases as the level of stochasticity increases while center correlation decreases. Error bar is STD from 6 RP and MFTMA seeds.

## SM 6.1 VOneNets

Figure SM18 replicates experiments from Figures 3B and 3C, except that all networks now receive the exact same perturbations drawn from random corners of the $L_\infty$ $\epsilon = 8/255$ sized ball around an image ($\tilde{x} = x + \text{sign}(\mathcal{N}(0, 1)\epsilon)$, demonstrating that while random perturbations cause generally less of an increase in capacity than FGSM-based perturbations, largely the trends are the same.

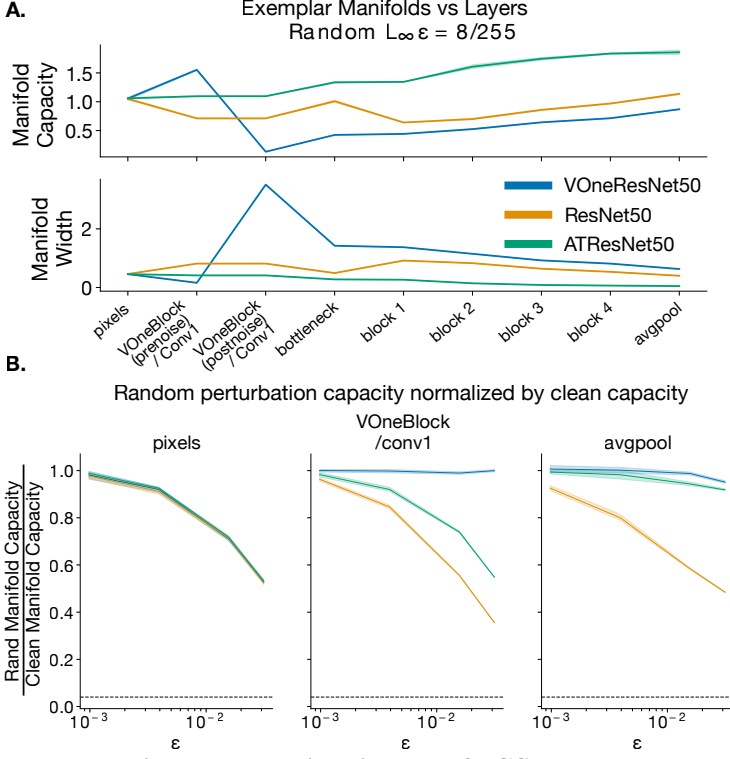

Figure SM18: **Random $\epsilon$-sized perturbations instead of FGSM-based perturbations for exemplar manifold analysis yields largely the same result** (**A**) Equivalent to Figure 3B but random corner perturbations are used instead of FGSM. (**B**) Equivalent to Figure 3C but random corner perturbations are used instead of FGSM. Error bars for all plots represent STD over 5 RP and MFTMA seeds.

## SM 6.2   Auditory Network

Auditory results conducted with adversarial exemplar manifolds (Figure 5) were replicated using random exemplar manifolds. Similar to the results with $L_\infty$ adversarial exemplar manifolds in Figure 5, we analyzed random exemplar manifolds constructed by taking points from the shell of the $L_\infty$ ball at $\epsilon = 10^{-3}$ (Figure SM19A). In the average pooling layer, we see that the capacity for random exemplar manifolds is similarly high for ATCochResNet50 and StochCochResNet50 without inference stochasticity, and that the capacity of StochCochResNet50 with inference stochasticity is lower than CochResNet50. We saw similar replications for the normalized capacity as a function of the $\epsilon$ (Figure SM19B). Taken together with the findings from VOneNet, these results suggest that the measured geometric differences between networks cannot be due to differences in the manifold sampling and instead reflect differences in the internal transformations.

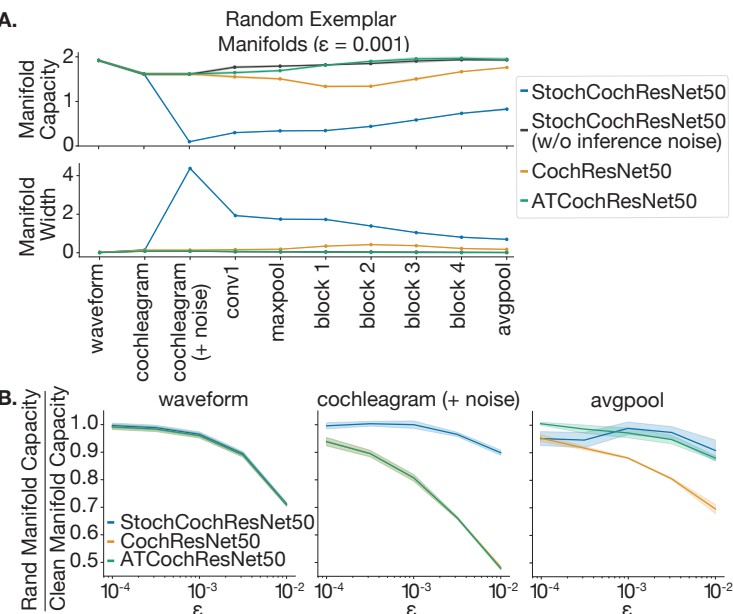

Figure SM19: **Random exemplar manifold geometry for auditory networks.** Replication of Figure 5 using random points from the shell of the $L_\infty$ ball at $\epsilon = 0.001$ to construct the exemplar manifold. (**A**) Capacity of random exemplar manifolds for layers of auditory models. For networks without stochasticity, the stochastic layer representation is equal to the cochleagram representation. Error bars are STD across 5 random projection seeds. (**B**) Random exemplar capacity normalized by the clean manifold capacity as a function of perturbation size. Error bars are STD across 5 random projection and MFTMA seeds. Data for StochCochResNet50, CochResNet50, and ATCochResNet50 is overlapping at waveform layer (as the same exemplar manifolds are tested for all networks) and data for CochResNet50 and ATCochResNet50 is overlapping at cochleagram layer, as this is also deterministic.

## SM 7   Variability of manifold widths and random projections

In our main paper, the error bars for class and exemplar manifold analysis represent the STD of different random projection (RP) seeds and MFTMA seeds. In particular, this means that we have already averaged out the variability across different manifold widths before computing the STD error bars. Note, unlike manifold width, the full manifold capacity measure is a systems level property, and to be computed requires averaging over the manifold geometric properties. Thus, we chose to include error bars for width that similarly demonstrate a systems level property. To give a sense of the general distribution of exemplar manifold widths, in Figure SM20 we show the raw distributions of exemplar manifold widths for VOneResNet50, GVOneResNet50, ResNet50, and ATResNet50 at the VOneBlock (post noise) / conv1 layer. We note that the distributions are well separated, and easily differentiable. In addition, we include Figure SM21, which plots the raw distributions after RP and MFTMA seeds for the same layer and exemplar stimuli. Variability of RP and MFTMA seeds is quite low and again networks easily separable.

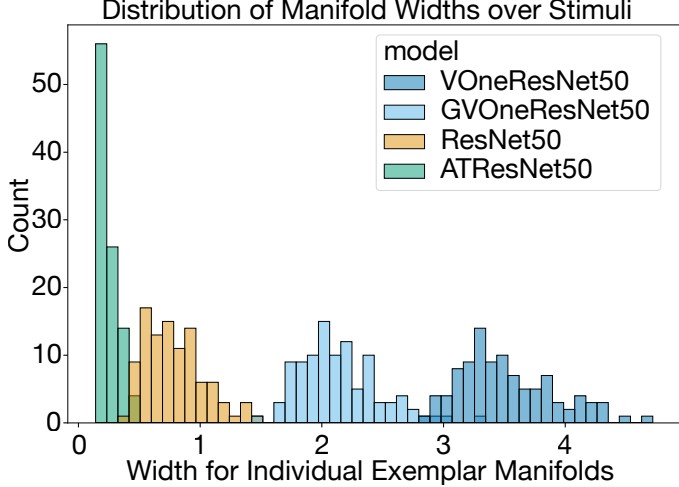

Figure SM20: **Raw distribution of exemplar manifold widths** The raw distribution of exemplar manifold widths for VOneResNet50, GVOneResNet50, ResNet50, and ATResNet50 at the VOneBlock (post noise) / conv1 layer show that manifold width for each network is reasonably well separated.

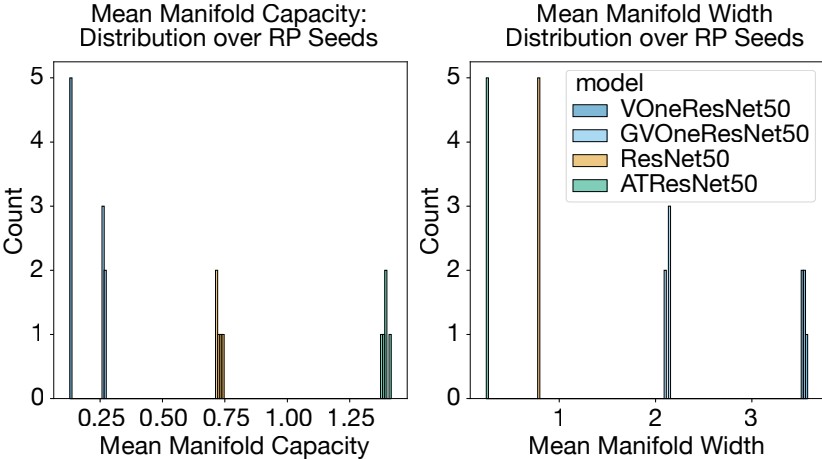

Figure SM21: **Raw distribution of RP and MFTMA seeds for exemplar manifold capacity and width** The raw distribution exemplar manifold capacity and width for 5 RP and MFTMA seeds for VOneResNet50, GVOneResNet50, ResNet50, and ATResNet50 at the VOneBlock (post noise) / conv1 layer show that variability from these sources is low and each network is well separated.

# SM 8  Pairwise Distribution Analysis

This section provides an additional way of looking at the overlap between clean and adversarial exemplar manifolds using a pairwise distribution analysis of the Gaussian CIFAR-VOneNet model. To give a geometric intuition of the measure, Figure SM22 shows the relative positions of two manifolds in the feature space and its corresponding pairwise distance distribution in different conditions. In Figure SM22A, the two manifolds are disjoint and both within-manifold distributions are not overlapped with the between-manifold distribution. In Figure SM22B, the two manifolds are partly overlapped and the within-manifold distribution is also partly overlapped with the between-manifold distribution. Finally, in Figure SM22C, when one of the manifolds is inclusive of the other, the mode of that within-manifold distribution will be larger than the mode of the between-manifold distribution. Figure SM23 shows the pairwise distance distribution of three layers after the stochastic `conv 1` `(postnoise)` layer, `block 1` layer and `avgpool` layer for low and high noise strength. A similar analysis is performed for the Poisson CIFAR-10 VOneNet in SM24.

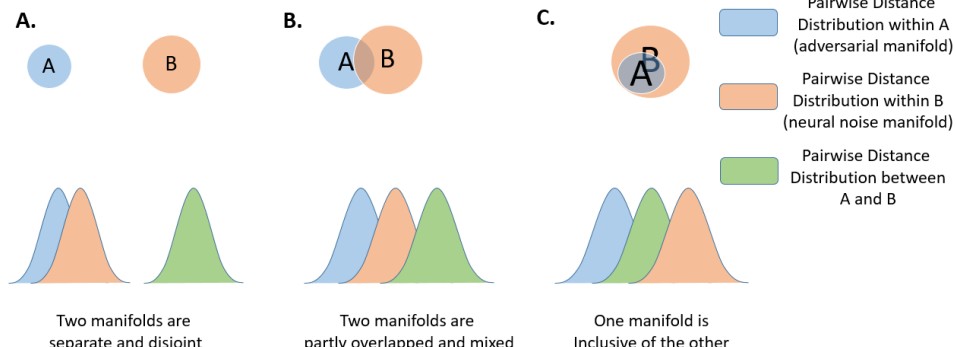

Figure SM22: **Illustration of pairwise distance analysis** The illustration shows the relative position of two clouds in the feature space and its corresponding pairwise distance distribution in three different scenarios: **(A)** two clouds are disjoint, **(B)** two clouds are partly overlapped and **(C)** one cloud is inclusive of the other. The more overlap between the between-manifold distribution and one of the within-manifold pairwise distance distributions, the more that these two clouds are overlapped in the feature space.

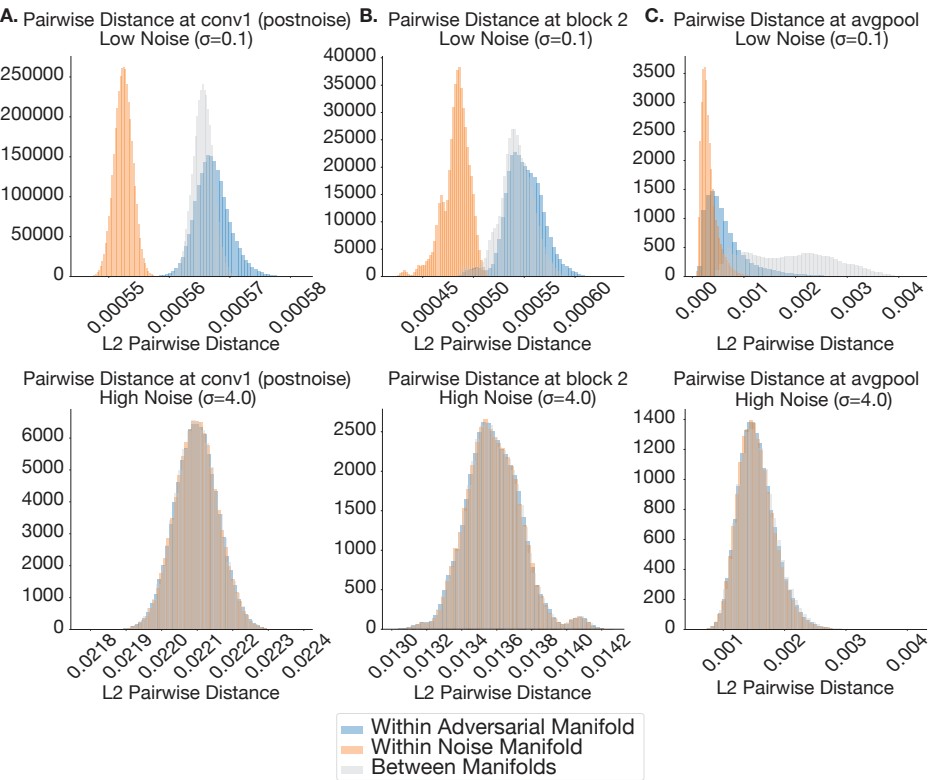

Figure SM23: **Gaussian noise model pairwise distance distributions** Pairwise distance distribution at layers **(A)** `conv1 (postnoise)`, **(B)** `block 2` and **(C)** `avgpool` of models with low noise level ($\sigma = 0.1$) and high noise level ($\sigma = 4.0$). At all measured layers, the adversarial exemplar and clean exemplar manifolds are more overlapped in the model with higher noise level ($\sigma = 4.0$)

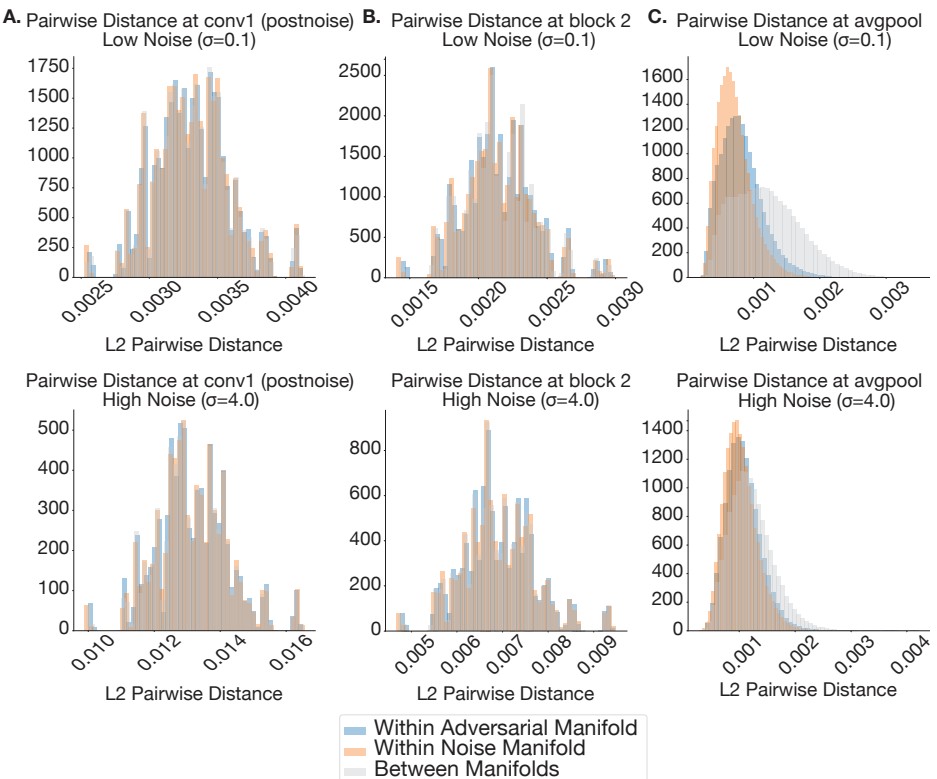

Figure SM24: **Pairwise distance distribution of Poisson-like noise model** Pairwise distance distribution at layers **(A)** conv1 (postnoise), **(B)** block 2 and **(C)** avgpool. At the early and intermediate layers (layer 2 and layer 4), the degree of manifold overlap is not visibly different between the low and high noise levels. However, at the late layer (layer 7), the model with high noise level has significantly more overlap between stochastic adversarial and clean manifolds than the model with lower levels of stochasticity. The level of stochasticity for the Poisson-like stochasticity model is defined by the variance/mean ratio. For standard Poisson, the variance/mean ratio is 1.