# OpenReview forum: "Neural Population Geometry Reveals the Role of Stochasticity in Robust Perception"
_NeurIPS.cc/2021/Conference — NeurIPS 2021 Poster_

### Official Review · Reviewer_UJnB · 2021-07-13

**Rating:** 8
**Confidence:** 4

**Summary:**

The authors note that stochasticity is an important component of recent attempts to improve network adversarial robustness by emulating known biological mechanisms. They also demonstrate their own similar result for a network trained on auditory signals. Motivated by this, they analyze the representations formed by a set of example networks to understand the relationship between representation geometry and robustness. Their analysis reveals differences among networks that underwent standard training, adversarial training, and those with stochasticity added. They claim that this analysis improves our understanding of mechanisms of robust perception as well as our understanding of the role stochasticity plays in biological computation.

**Limitations And Societal Impact:**

yes

**Main Review:**

*Review summary:*

The MFTMA analysis method reveals clear differences among the networks tested. Specifically, I find it compelling how clearly the authors demonstrated that two strategies for achieving robustness can show opposing behavior in terms of their analysis. I note a bunch of nitpick concerns in my review below, but overall I found the paper to be well written and the results interesting. I am recommending the paper for acceptance, and will further increase my score if my concerns are addressed.

*Strengths:*


The paper is well written and has helpful extended explanations in the appendix. Their analysis provides a very interesting perspective on how two different methods for achieving robustness (adversarial training vs noise training) can lead to alternative mechanisms. Their controls, both within the main text and in the appendix, are thorough and convincing of the overall result.

*Weaknesses:*
1) Claims:
1.a) I think the paper falls short of the high-level contributions claimed in the last sentence of the abstract. As the authors note in the background section, there are a number of published works that demonstrate the tradeoffs between clean accuracy, training with noise perturbations, and adversarial robustness. Many of these, especially Dapello et al., note the relevance with respect to stochasticity in the brain. I do not see how their additional analysis sheds new light on the mechanisms of robust perception or provides a better understanding of the role stochasticity plays in biological computation. To be clear - I think the paper is certainly worthy of publication and makes notable contributions. Just not all of the ones claimed in that sentence.

1.b) The authors note on lines 241-243 that “the two geometric properties show a similar dependence for the auditory (Figure 4A) and visual (Figure 4B) networks when varying the eps-sized perturbations used to construct the class manifolds.” I do not see this from the plots. I would agree that there is a shared general upward trend, but I do not agree that 4A and 4B show “similar dependence” between the variables measured. If nothing else, the authors should be more precise when describing the similarities.

2) Clarifications:
2.a) The authors say on lines 80-82 that the center correlation was not insightful for discriminating model defenses, but then use that metric in figure 4 A&B. I’m wondering why they found it useful here and not elsewhere? Or what they meant by the statement on lines 80-82.

2.b) On lines 182-183 the authors note measuring manifold capacity for unperturbed images, i.e. clean exemplar manifolds. Earlier they state that the exemplar manifolds are constructed using either adversarial perturbations or from stochasticity of the network. So I’m wondering how one constructs images for a clean exemplar manifold for a non-stochastic network? Or put another way, how is the denominator of figure 2.c computed for the ResNet50 & ATResNet50 networks?

2.c) The authors report mean capacity and width in figure 2. I think this is the mean across examples as well as across seeds. Is the STD also computed across examples and seeds? The figure caption says it is only computed across seeds. Is there a lot of variability across examples?

2.d) I am unsure why there would be a gap between the orange and blue/green lines at the minimum strength perturbation for the avgpool subplot in figure 2.c. At the minimum strength perturbation, by definition, the vertical axis should have a value of 1, right? And indeed in earlier layers at this same perturbation strength the capacities are equal. So why does the ResNet50 lose so much capacity for the same perturbation size from conv1 to avgpool? It would also be helpful if the authors commented on the switch in ordering for ATResNet and the stochastic networks between the middle and right subplots.

3) General curiosities (low priority):
3.a) What sort of variability is there in the results with the chosen random projection matrix? I think one could construct pathological projection matrices that skews the MFTMA capacity and width scores. These are probably unlikely with random projections, but it would still be helpful to see resilience of the metric to the choice of random projection. I might have missed this in the appendix, though.

3.b) There appears to be a pretty big difference in the overall trends of the networks when computing the class manifolds vs exemplar manifolds. Specifically, I think the claims made on lines 191-192 are much better supported by Figure 1 than Figure 2. I would be interested to hear what the authors think in general (i.e. at a high/discussion level) about how we should interpret the class vs exemplar manifold experiments.

4) Nitpick, typos (lowest priority):
4.a) The authors note on line 208 that “Unlike VOneNets, the architecture maintains the conv-relu-maxpool before the first residual block, on the grounds that the cochleagram models the ear rather than the primary auditory cortex.” I do not understand this justification. Any network transforming input signals (auditory or visual) would have to model an entire sensory pathway, from raw input signal to classification. I understand that VOneNets ignore all of the visual processing that occurs before V1. I do not see how this justifies adding the extra layer to the auditory network.

4.b) It is not clear why the authors chose a line plot in figure 4c. Is the trend as one increases depth actually linear? From the plot it appears as though the capacity was only measured at the ‘waveform’ and ‘avgpool’ depths; were there intermediate points measured as well? It would be helpful if they clarified this, or used a scatter/bar plot if there were indeed only two points measured per network type.

4.c) I am curious why there was a switch to reporting SEM instead of STD for figures 5 & 6.

4.c) I found typos on lines 104, 169, and the fig 5 caption (“10 image and”).

**Time Spent Reviewing:**

4.5

---

> ### Author Response · Authors · 2021-08-10
> **Response to reviewer UJnB**
>
> We thank the reviewer for the time and effort spent on this review. We were pleased the reviewer found our work to be well written, interesting, and thorough. We appreciate the feedback offered by the reviewer, and believe that incorporating changes to address these questions and critiques will improve our paper. In particular:
>
> 1.a) We take the reviewers point that the claims in the last line of our abstract are too broad. We propose changing this line to the following: “Our results shed light on the strategies of robust perception utilized by adversarially trained and stochastic networks, and help explain how stochasticity may be beneficial to machine and biological computation.” We believe these claims are warranted because prior to our work it was unclear how stochasticity helped with robustness in the VOneNet, and whether or not the mechanism of adversarial robustness between adversarially trained networks and VOneNets were the same. If the reviewer still takes issues with these claims, we are open to suggestions for further change.
>
> 1.b) We largely agree that the lines in 241-243 lack precision. We will replace this with a more careful version of the interpretation. A draft for lines 241-246 is the following: Figure 4A and 4B show that in both visual and auditory networks an increase in size of the adversarial perturbation leads to an increase in manifold width and center correlation (two key variables leading to the capacity). While this general trend is present for all networks, the networks that are more adversarially robust have less change across both metrics as the perturbation size increases. This provides further evidence that the manifold metrics are a useful way to interpret the internal geometries that lead to adversarial robustness, and points to the same underlying mechanism in visual and auditory modalities for the classification degradation in the presence of adversarial vulnerability -- specifically that the class manifolds become larger and more correlated when adversarial perturbations are present.
>
> 2.a) Thank you for pointing out the inconsistency in lines 80-82. In most cases, we felt the trends we were interested in highlighting were clear and strong from manifold width alone, and for space reasons chose to focus on those whenever possible. In Figure 4a and 4b we wanted to give a full characterization of the geometry for comparison between the auditory and visual networks, and thus included correlation. In retrospect we agree the statement on 80-82 is misleading, and we will adjust the language in the final version to give a more accurate account of our choices.
>
> 2.b) The clean exemplar manifolds for networks without stochasticity are defined as having the maximum capacity (essentially, treating the manifold as a single point), which is equal to 2 (Chung et al. 2018). We will clarify this in the text.
>
> 2.c) In Figure 2, because capacity is a collective systems-level property which is characteristic of all the manifolds being measured, the error bars are only showing the standard deviation over different random projections. On the other hand, because the manifold width is a lower level property which can be computed on a per manifold basis, the error bars are showing the standard deviation across 5 random projections and all 100 image exemplar manifolds. In terms of variability across images, we will add a supplementary figure showing the distribution of exemplar manifold widths (https://imgur.com/a/eMadfaX), which directly shows the variation of manifold width for different images.
>
> 2.d) On figure 2c we do not include a perturbation size of 0 which would have a value of 1, as observed by the reviewer. Rather the smallest attack that we consider on the plot is $\epsilon=10^{-3}$. Even for this small perturbation size the capacity for the regular ResNet becomes degraded, leading to the drop in capacity by the average pooling layer, while in earlier layers the effect is less noticeable. We believe the differences that arise between conv1 and avgpool are due to the different learned representations of each network -- some learned representations may be able to mitigate the adversarial perturbations and maintain high capacity (ie what you see in the ATResNet) while others may essentially amplify the perturbations leading to a decrease in accuracy and performance (as observed in standard ResNet50). The switching in the order of ATResNet and VOneResNet between the early and late layers of networks indicates that the VOneResNet is more stable to increased adversarial perturbation sizes in the early layers but loses capacity more rapidly in the later layers as the perturbation size increases, whereas the ATResNet is not as stable in the early layers but makes up for it by the later layers, further illustrating the differences between these two defenses.
>
> 3.a) Error bars are computed across random projections, to capture some of this variability. In general, the choice of random projection does not lead to much variability as long as the dimension for the projection is high. This is because random projections can preserve the geometry of data to a surprising level of accuracy, as demonstrated in theoretical work (Johnson and Lindenstrauss 1984, Candes and Tao 2005). We also propose adding a supplemental figure (https://imgur.com/a/duUehsh) showing the distribution from random projection seeds.
>
> 3.b) The class manifold experiments are most tightly coupled to the accuracy of the network on the classification task. Class manifolds have variability due to differences in exemplars in the class in addition to the stochasticity or adversarial perturbations, while exemplar manifolds only have variability due to stochasticity or adversarial perturbations. Thus, multiple types of exemplar manifold geometries could lead to the same class manifold capacity and width. The schematic included here (https://imgur.com/a/0ZblZAt ) may provide some intuition, and we will add this as a supplemental figure. For networks analyzed in the paper, at the average pooling layer, both the ATResNet50 and the VOneResNet50 have high class manifold capacity when manifolds are constructed from adversarially perturbed stimuli (Figure 1 C, D), however the exemplar manifold experiments in Figure 2B suggest that the manifolds are constructed in different ways, such that the ATResNet50 has high exemplar manifold capacity at the average pooling layer while the VOneResNet50 has much lower exemplar manifold capacity.
>
> 4.a) The question of which artificial neural network architecture best represents the human brain is an interesting and open question, and we acknowledge that some of the choices for architecture explored here are somewhat arbitrary and determined based on availability of previous models and limited time and computational resources. As the reviewer notes, the VOneNet architecture ignores processing that occurs before V1, and swaps out the first set of convolutional layers for ones that are based on V1 responses, while maintaining the image input representation. Past work using cochleagrams has typically swapped in a cochleagram for the image representation, essentially including hard-coded biological filters before any additional processing (Kell et al. 2018). In networks with cochleagram inputs, the first layer of convolutional filters tends to be interpreted as a learned set of spectrotemporal filters, akin to the spectrotemporal filters that are present within the primary auditory cortex. More practically, this first conv-relu-maxpool layer also performs a large amount of downsampling, which we wanted to maintain before feeding the cochleagram into the first residual block. It also expands the channel dimensions from 1 to 96. This is the rationale for line 208, and we will attempt to better summarize the nuances about model architecture using extra space in our final paper.
>
> 4.b) We originally wanted to highlight the difference in capacity at the avgpool layer, but agree that using a line plot was confusing, as was also pointed out by Reviewer fXyv. We will update the Figure 4c plot to include more layers (see https://imgur.com/a/SXsPYwR), where you can see that the trend is not linear between the waveform and avgpool layer. We left this as a line plot as (in our opinion) it becomes a bit difficult to read as a scatter or bar, however we acknowledge the linear interpolations between measured layers are not guaranteed.
>
> 4.c) Thank you for pointing out the error bar changes and typos. All error bars will be changed to STD in the final paper, and we will fix the typos.
>
>
> References:
>
> [1] Chung, S., Lee, D. D., & Sompolinsky, H. (2018). Classification and Geometry of General Perceptual Manifolds. Physical Review X, 8(3), 031003.
>
> [2] Johnson, W. B., & Lindenstrauss, J. (1984). Extensions of Lipschitz mappings into a Hilbert space 26. Contemporary mathematics, 26.
>
> [3] Candes, E. J., & Tao, T. (2005). Decoding by linear programming. IEEE transactions on information theory, 51(12), 4203-4215.
>
> [4] Kell, A. J., Yamins, D. L., Shook, E. N., Norman-Haignere, S. V., & McDermott, J. H. (2018). A task-optimized neural network replicates human auditory behavior, predicts brain responses, and reveals a cortical processing hierarchy. Neuron, 98(3), 630-644.

---

> > ### Comment · Reviewer_UJnB · 2021-08-19
> > **Concerns sufficiently addressed**
> >
> > Thank you for revising the text & creating the extra supplementary plots. In light of these revisions as well as the ones provided for the other reviewers, I have decided to increase my score.

---

### Official Review · Reviewer_8wmG · 2021-07-16

**Rating:** 7
**Confidence:** 3

**Summary:**

It is well known that a deep learning model is fragile to adversarial perturbation, whereas a biological system is not. This study investigate the geometry of internal representations (especially, in the penultimate later) of a DL model, that trained using adversarial examples, and that with biologically-inspired component. The basic approach is based on what the authors call MFTMA.


**Main Review:**

Roughly speaking, the reviewer failed in finding a significance in the results and discussions.

(1) The relationship with references [20,22] are unclear. The word "MFTMA" does not appear in the references. The supplementary material 2.1 suggests that it is a synonym fot the method in [8]. Please discuss the relationship with references.

(2) The references [20,22] demonstrated the manifold size is related to the accuracy. Then, the contribution of this study is unclear. All experiments can be done using the accuracy as the measure instead of the manifold size. The only thing that the reviewer considers as the contribution is the suggestion that the manifold analysis is applicable to adversarial perturbation. If so, the introduction is inappropriate.

(3) According to Figure 1, the top-1 accuracy is proportional to the manifold capacity. A vanilla ResNet loses its accuracy and manifold capacity with a strong attack, whereas a ResNet trained with adversarial perturbation does not. VOneResMet is more robust to attack than the vanilla. The similar results are found in auditory data recognition in Section 4.

Several previous works such as Miyato et al. demonstrated that all the dropout, random perturbation, and adversarial training smooth the score function. While dropout and random perturbation are stochastic and inefficient compared to adversarial training, their contribution is similar to adversarial training. Hence, it is unsurprising that "there exists an overlap between the stochastic representations of clean and adversarially perturbed stimuli".

Even through the very first paper by Goodfellow et al. suggested that the dropout is insufficient for robustness to the adversarial perturbation, many previous studies such as Dhillon et al. have attempted to improve the robustness by stochastic components. Hence, it is unsurprising that "stochastic representations provide a defense against adversarial perturbations."

- Miyato et al., "Virtual Adversarial Training: A Regularization Method for Supervised and Semi-Supervised Learning," IEEE TPAMI, 2019.
- Goodfellow et al., "Explaining and Harnessing Adversarial Examples," ICLR 2015.
- Dhillon et al., "Stochastic Activation Pruning for Robust Adversarial Defense," ICLR2018.


**Time Spent Reviewing:**

7

---

> ### Author Response · Authors · 2021-08-10
> **Response to reviewer 8wmG**
>
> We thank the reviewer for their time spent reviewing our paper. We hope that by clarifying key points in our paper and addressing concerns raised by the reviewer, we will be able to convince the reviewer that our work is novel and significant, as noted by other reviewers. Specifically:
>
> (1) We appreciate the reviewer pointing out that MFTMA as a term was not directly used in our cited works. We apologize for this confusion, and we will clarify this ambiguity in our final work. In short, Mean-Field theoretic manifold analysis (MFTMA) refers directly to the method introduced in [20,22]. This shorthand for the method was first used in “Untangling in Invariant Speech Recognition” by Stevenson et al. NeurIPS 2019, and the provided code formed the basis for our analysis as noted in the supplement. We will update the main text to include this citation as well.
>
> (2) It is incorrect to say that  “all experiments can be done using the accuracy as the measure instead of the manifold size”. While the class manifold capacity in the penultimate layer is indeed related to classification accuracy (for instance, as shown in Supplementary Figure 10), the manifold metric allows us to investigate how class manifolds are untangled as they travel through all layers of the network. Critically, accuracy alone is insufficient to distinguish between the different mechanisms of robustness that we found -- while VOneNet and ATResNet50 have comparable accuracy, manifold analysis of exemplar manifolds reveals stark differences in the internal representations of these two networks, signifying distinct mechanisms to achieve robust accuracy. Furthermore, recent work has discussed how simple analysis with linear classifiers is limited (Hewitt and Liang, 2019), and that more structural studies are needed for investigating the internal layers of networks (Ivanova et al, 2021), and we believe our study situates the analysis of robustness mechanisms into the developing field of neural population geometry (Kriegeskorte and Kievit, 2013; Chung and Abbott, 2021)
>
> (3) Regarding significance of our results with respect to previous work, we believe the reviewer may have interpreted lines 258-259 in a way other than we intended. We agree with the reviewer’s summary that stochastic activations have been previously explored; however, previous works do not investigate whether the defense mechanisms introduced through stochastic defenses such as that in Dhillon et al. or Miyato et al. yield the same underlying changes in representational structure as adversarial training. We do not intend to claim that we have discovered a new type of adversarial defense; the major contribution of our work is to characterize how stochastic representations change the underlying representational geometry in networks in a way that is beneficial for adversarial robustness. We note that it is practically useful to consider these differences, as it may offer insight into when one adversarial defense method can benefit from being combined with another adversarial defense method, or when ostensibly different defenses are redundant. As mentioned in our response to Reviewer fXyv, we will add to the introduction to more clearly place the work in the context of Dapello et al. 2020, where it was unclear whether the mechanism for robustness in VOneNets was similar to adversarial training. We will clarify the summary statement in lines 258-259 to better capture the results from Sections 3-4.
>
> References
>
> [1] Hewitt, J., & Liang, P. (2019). Designing and interpreting probes with control tasks. arXiv preprint arXiv:1909.03368.
>
> [2] Ivanova, A. A., Hewitt, J., & Zaslavsky, N. (2021). Probing artificial neural networks: insights from neuroscience. arXiv preprint arXiv:2104.08197.
>
> [3] Chung, S., & Abbott, L. F. (2021). Neural population geometry: An approach for understanding biological and artificial neural networks. arXiv preprint arXiv:2104.07059.
>
> [4] Kriegeskorte, N., & Kievit, R. A. (2013). Representational geometry: integrating cognition, computation, and the brain. Trends in cognitive sciences, 17(8), 401-412.

---

> > ### Comment · Reviewer_8wmG · 2021-08-18
> > **Comments**
> >
> > Thank you for response.
> >
> > > the manifold metric allows us to investigate how class manifolds are untangled as they travel through all layers of the network. Critically, accuracy alone is insufficient to distinguish between the different mechanisms of robustness that we found
> >
> > I see. The accuracy is obtained only at the last layer. However,  the manifold analysis can be done also in penultimate layers quantitatively, and it can evaluate the contribution of each layer. Is my understanding correct?
> >
> > If so, I think that your study contribute to the network design, not only to the analysis.

---

> > > ### Author Response · Authors · 2021-08-18
> > > **Response to Reviewer's Comments**
> > >
> > > The reviewer is correct -- MFTMA allows us to quantitatively evaluate the effect of each layer on the separation of class and exemplar manifolds. For an example of this, we encourage the reviewer to review Figure 2B for vision networks, or the updated Figure 4C https://imgur.com/a/SXsPYwR for auditory networks.
> > >
> > > We further agree that our study also contributes to network design -- by analyzing the VOneBlock in visual networks, we are able to successfully suggest similar changes to auditory networks.
> > >
> > > Thank you for the feedback. We will incorporate it into the final draft.

---

> > > > ### Comment · Reviewer_8wmG · 2021-08-19
> > > > **Acknowledgments**
> > > >
> > > > > we are able to successfully suggest similar changes to auditory networks.
> > > >
> > > > This is the second bullet in the bottom of introduction and Section 4.
> > > >
> > > > Thank you very much for your detailed responses, and sorry for my limited understanding. I *corrected* my score to 7.

---

### Official Review · Reviewer_fXyv · 2021-07-16

**Rating:** 7
**Confidence:** 3

**Summary:**

The paper compares networks that are adversarially trained with networks that have biologically inspired stochasticity in the first layer, including previous visual object-recognition networks and a new auditory network. The analysis is based on manifolds of classes and exemplars in representation space, and considers measures of manifold size and overlap. Robustness to adversarial attacks is also considered. The analysis highlights different responses of the stochastic and adversarially trained networks to adversarial attacks.

**Ethical Concerns:**

The work does not raise ethical concerns.

**Limitations And Societal Impact:**

The work has a minor societal impact.

**Main Review:**

Originality:
This is a new kind of analysis, and I don’t think it has been used to study adversarial robustness before. It makes sense to do so.

Quality:
The work is technically sound and thorough.

In comparing visual networks, the paper varies two things at once. The paper focuses on the fact that VOneResNet50 has stochastic responses while ATResNet50 has adversarial training. But another difference is that VOneResNet50 has biologically inspired first-layer filters, while ATResNet50 has standard first-layer filters. It seems cleaner to either address the first difference in isolation, or to test all combinations. The authors leave this for future work (line 343) but it may be a limitation.

Clarity:
The writing is clear and engaging.

Significance:
Here I am uncertain. The results with undefended and adversarially trained networks are unsurprising. It is interesting that stochasticity widens exemplar manifolds in early layers, contrary to adversarial training, but I am not sure what else I would have expected. I may need more guidance from the authors about what to take from the study.

The last section of the paper shows that exemplar manifolds due to stochasticity and adversarial perturbations overlap, less so at high attack strengths, and more so with high stochasticity. Here again, I am not sure what else might have been expected.

Minor:
“basis of this stochasticity and its implications for information processing are open questions in neuroscience [19]” The reference is good but not recent.

In Figure 2B, I am not clear about the significance of the decreasing manifold width in ATResNet50, versus the fact that for other networks, width is higher at the output than the pixels. I don’t think it matters, but maybe the authors could discuss this in more detail, or add a plot of un-perturbed manifold widths for additional context.

Another helpful point of context would be the accuracy vs. attack strength added to Figure 1C,D.

In Figure 4C, the black line is easy to miss. Lines may not be the best representation in this plot because the ordinate isn’t continuous, and multiple levels are skipped.

I missed which layers are analyzed in Figure 5.


**Time Spent Reviewing:**

5 hours

---

> ### Author Response · Authors · 2021-08-10
> **Response to reviewer fXyv**
>
> We thank the reviewer for taking the time to review our paper, and were happy to find that the reviewer found our work to be original, technically sound, and clearly presented. Below we address the reviewer’s concerns and where appropriate make suggestions for how we will improve the paper in response:
> * We acknowledge the reviewer’s concern that by varying two things at once (the stochasticity of the responses and the presence of biological filters) we are potentially confounding two effects. To address this concern, we will include this additional control figure (https://imgur.com/a/ec9I6AR), showing a comparison of VOneNet without stochasticity, as well as a standard ResNet with stochastic activations, demonstrating that the stochasticity rather than the biological filters is the major driving factor for the differences in the manifold geometry. We will also rephrase lines 341-343 to incorporate this analysis.
> * Regarding the significance of our work, we believe that more clearly placing our work in the context of Dapello et al. (2020) will make the significance more clear. Dapello et al. found that compared to standard networks, both adversarially trained networks and VOneNets had improved adversarial robustness and also greater similarity to the primate primary visual cortex. However, it remained unclear why VOneNets were more robust, and in particular if the mechanism of robustness was similar to adversarial training. Here we demonstrate that there are multiple mechanisms at play, which suggests that in future work adversarial training could be stacked with the VOneBlock to further improve robustness. We appreciate the reviewer raising this concern, and will make these implications clear in the final draft of our paper.
>
> Addressing minor points:
> * We agree that the reference for stochasticity in neuroscience is outdated, though we have had difficulty finding more recent reviews. We propose to also cite Goris et al., 2014, and we would happily include any other references the reviewer is able to suggest.
> * In Figure 2B, the decreasing manifold width in ATResNet50 indicates that as the exemplar manifolds propagate through the adversarially trained network, they are generally shrunk down to smaller and smaller volumes, indicating greater stability of this network with respect to local image perturbations. Meanwhile, in the undefended ResNet50, the width of the exemplar manifolds generally increases, indicating that the network is more sensitive to adversarial perturbations than the ATResNet50 and later stages of the networks may amplify the perturbations rather than shrinking them.
> * Accuracy vs. attack strength is presented for VOneResNet50 and VOneResNet50 in Supplementary Figure 1. We will add the other models (ATResNet50 and ResNet50) to this Figure for comparison, and if space allows will move it to the main text.
> * We accept the reviewers critique of the visualization in 4C and will improve this for the final paper (it was also mentioned by reviewer UJnB). We opted to increase the number of layers presented on this figure, while maintaining it as a line plot. This updated figure can be found here: https://imgur.com/a/SXsPYwR.
> * Figure 5 analyzes the stochastic representations of the network, which are the output of the VOneBlock for the CIFAR and ImageNet models, and the stochastic cochleagram for the auditory model. We will clarify which layers are analyzed in the text and figure caption.
>
> References
>
> [1] Dapello, J., Marques, T., Schrimpf, M., Geiger, F., Cox, D. D., & DiCarlo, J. J. (2020). Simulating a Primary Visual Cortex at the Front of CNNs Improves Robustness to Image Perturbations. Advances in Neural Information Processing Systems 33 (NeurIPS 2020).
>
> [2] Goris, R. L. T., Movshon, J. A., & Simoncelli, E. P. (2014). Partitioning neuronal variability. Nature Neuroscience, 17(6), 858–865.

---

> > ### Comment · Reviewer_fXyv · 2021-08-18
> > **Satisfied with authors' responses**
> >
> > Thank you for addressing my comments. I support acceptance of the paper.
> >
> > I don't know if I can suggest a more recent review of the "basis of stochasticity and its implications for information processing in neuroscience". If you think it's appropriate, a couple of recent papers with diverse perspectives could also be cited to support the idea that the question remains open, e.g.:
> >
> > [1] R. Echeveste, L. Aitchison, G. Hennequin, and M. Lengyel, 2020, “Cortical-like dynamics in recurrent circuits optimized for sampling-based probabilistic inference,” Nat. Neurosci., vol. 23.
> > [2] Z. W. Davis, L. Muller, J. Martinez-trujillo, T. Sejnowski, and J. H. Reynolds, 2020, “Spontaneous travelling cortical waves gate perception in behaving primates,” Nature, vol. 587.

---

### Official Review · Reviewer_D6LM · 2021-07-16

**Rating:** 7
**Confidence:** 3

**Summary:**

The submission investigates the role of response stochasticity, a feature of biological neurons, in robustness to adversarial attacks in deep neural networks.  They use manifold analysis to compare the neural population geometry of standard-trained, adversarial-trained and stochastic networks and find geometrical signatures of the different conditions. They show that the robustness achieved through adversarial training is qualitatively different than that achieved by the stochastic network. They characterize how increasing noise affects the resulting manifolds. They observe a tradeoff between adversarial robustness and classification performance and can explain this tradeoff in terms of geometric features of the representations. These results (on images) are replicated in the auditory domain.

**Limitations And Societal Impact:**

The authors clearly acknowledge the limitations of their analysis. They do not mention any potential negative societal impact but none comes to mind.

**Main Review:**

The submission employs recently developed manifold analysis techniques and sheds new light on a previously documented phenomenon. The demonstration of the value of stochasticity in auditory models is novel.

MFTMA is an exciting analysis technique which was proposed a few years ago but has yet to be widely employed. I am pleased to see it used here where is has helped to shed light on an interesting phenomenon. The work is complete and the authors clearly identify avenues for future work which are not addressed in the present submission.

In general, the submission is clearly written. However, there may be room for improvement in the introduction + related work section (often two sections, but here combined into one). In particular, I found the logical jump from paragraph 2 to 3 unclear. The particular experimental design and methodology does not feel clearly motivated in the text (although I believe that it is motivated). I imagine the authors have tried to condense this first section to make room for the rest of the paper but it feels slightly too condensed.

minor points:
- In the first paragraph the authors write "establishing them as leading candidate models of human perception". "Human perception" denotes a much broader set of capacities than those for which we have validated neural network models. Consider scaling back to "visual object recognition".
- line 269: "We include investigation how the level of noise...". Are there some missing words here?

The main conclusions are novel. Some of the results replicate previous findings. The work constitutes progress towards understanding the role of stochasticity in neural computation, which has potential implications for both neuroscience and machine learning. It also constitutes a step towards biologically-inspired deep learning.

**Time Spent Reviewing:**

2.25

---

> ### Author Response · Authors · 2021-08-10
> **Response to reviewer D6LM**
>
> We thank the reviewer for their time spent reviewing our paper; we are pleased to hear that they found our work novel and interesting, and we will happily incorporate their suggestions to make our paper stronger.
>
> In particular:
> * We agree that “there may be room for improvement in the introduction + related work section,” and if accepted we will make use of our additional space to provide more background, and in particular give a stronger motivation for our methodology and discussion of the significance of our work.
> * We accept that "establishing them as leading candidate models of human perception" is too strong a statement by itself. We used this phrasing in an attempt to acknowledge that we are investigating not just models of vision but also models of audition. We propose to qualify the claim as "establishing them as leading candidate models for several domains of human perception" and supplying references for each domain we are aware of, including visual object recognition (Rajalingham et al. 2018) and human speech and music perception (Kell et al. 2018).
> * We appreciate the reviewer pointing out the errors on line 269, and will change this to “We investigate how the level of noise changes...”
>
>
> References
>
> [1] Rajalingham, R., Issa, E. B., Bashivan, P., Kar, K., Schmidt, K., & DiCarlo, J. J. (2018). Large-Scale, High-Resolution Comparison of the Core Visual Object Recognition Behavior of Humans, Monkeys, and State-of-the-Art Deep Artificial Neural Networks. The Journal of Neuroscience: The Official Journal of the Society for Neuroscience, 38(33), 7255–7269.
>
> [2] Kell, A. J., Yamins, D. L., Shook, E. N., Norman-Haignere, S. V., & McDermott, J. H. (2018). A task-optimized neural network replicates human auditory behavior, predicts brain responses, and reveals a cortical processing hierarchy. Neuron, 98(3), 630-644.

---

### Decision · Program_Chairs · 2021-09-27

**Decision:**

Accept (Poster)

**Comment:**

This is an excellent paper with good consensus and engagement among the reviewers for the acceptance. It derives novel insights for mechanisms of robust perception, and will likely inspire researchers in the fields of adversarial robustness, biologically-inspired neural networks, and computational neuroscience, all important and active sub-fields in NeurIPS conference.